# Pocket MUSE: an affordable, versatile and high-performance fluorescence microscope using a smartphone

Yehe Liu [1], Andrew M. Rollins[1], Richard M. Levenson [2], Farzad Fereidouni[2] & Michael W. Jenkins[1,3]✉

Smartphone microscopes can be useful tools for a broad range of imaging applications. This manuscript demonstrates the first practical implementation of Microscopy with Ultraviolet Surface Excitation (MUSE) in a compact smartphone microscope called Pocket MUSE, resulting in a remarkably effective design. Fabricated with parts from consumer electronics that are readily available at low cost, the small optical module attaches directly over the rear lens in a smartphone. It enables high-quality multichannel fluorescence microscopy with submicron resolution over a 10× equivalent field of view. In addition to the novel optical configuration, Pocket MUSE is compatible with a series of simple, portable, and user-friendly sample preparation strategies that can be directly implemented for various microscopy applications for point-of-care diagnostics, at-home health monitoring, plant biology, STEM education, environmental studies, etc.

[1] Department of Biomedical Engineering, Case Western Reserve University, Cleveland, OH, USA. [2] Department of Pathology and Laboratory Medicine, University of California Davis Medical Center, Sacramento, CA, USA. [3] Department of Pediatrics, Case Western Reserve University, Cleveland, OH, USA. ✉email: wmj5@case.edu

Equipped with high-performance digital cameras, modern smartphones provide a highly accessible platform for advanced imaging tasks such as optical microscopy. With small footprints, low price-points, and built-in image processing capabilities, smartphone microscopes have shown utility when access to benchtop microscopes is limited. From point-of-care diagnostics[1–4] to education[5,6], smartphone-based microscopes have been developed for various applications for increasingly broader user groups. However, current smartphone microscope designs are often limited by significant trade-offs between cost, imaging performance, and functionality. The ability of smartphones to rapidly image specimens outside the lab is mitigated by complicated sample preparation that is typically needed for microscopy imaging (e.g., generation of conventional thin sections). To address these limitations, we introduce pocket microscopy with ultraviolet surface excitation (Pocket MUSE): an ultra-compact smartphone fluorescence microscope design that features versatility, excellent imaging performance, simplicity, and low cost.

Two major strategies for designing a smartphone microscope involve either multiple optical elements or only a single lens. When imaging performance and advanced functionalities are required, it is common to attach a smartphone to a designated system with multiple optical elements or even onto a standard optical microscope[7–9]. This type of design essentially provides the full capabilities of a conventional benchtop microscope, but often results in high cost and design complexity. Alternatively, one can implement a single positive lens with a short focal length immediately in front of the smartphone camera. However, imaging performance (e.g., magnification, resolution, etc.) is limited by the quality of the lens (e.g., aberrations). It is also difficult to incorporate additional components in the optical path needed for functionalities such as epifluorescence microscopy.

Using a single lens is a popular starting point for new smartphone microscope designs[6,10–12] because cost efficiency is becoming increasingly important especially to applications in low-resource settings. While maintaining a low cost and small footprint, several optically advanced design concepts have been developed to improve the performance and functionality of the more compact configurations. For instance, using a reversed smartphone camera lens as the objective helped to reduce optical aberrations, increase the effective field of view (FOV) while maintaining system cost[10]. To facilitate fluorescence functionality, colored polymer lenses helped to replace bulky and expensive filters[11]. Still, optical components are not the only major source of cost in smartphone microscope designs. Other mechanical tasks (e.g., focusing, positioning, etc.) can also add cost and structural complexity to the system. Advancements are continually improving the cost, compactness, and performance of smartphone microscopes.

Sample preparation is another essential component of microscopic imaging and can be responsible for bottlenecks in the deployment of smartphone microscopy. Like most conventional wide-field microscopes, smartphone-based systems are typically configured to image flat, thin, and stained samples[1,7,8,10,13]. However, when smartphone-based systems are more advantageous, access to conventional microscopy sample processing resources (e.g., fixation, embedding, microtoming) can be limited. One way to address this problem is to develop specialized sample holding and/or staining systems (e.g., microfluidic chambers[3,14]) to simplify the sample processing procedures. However, these solutions are often highly sample-specific. Thus, it would be beneficial to find simple and versatile sample preparation techniques for mobile microscopy applications without sacrificing the promise of the wide applicability of smartphone microscopes.

Here, we identified microscopy with ultraviolet surface excitation (MUSE)[15–18] as an effective complement to smartphone microscopes. Originally, MUSE was demonstrated as a promising tool for histopathology on benchtop microscopes. However, we found that adding MUSE functionality to a smartphone microscope can be highly synergistic in several different ways. Because sub-285 nm UV is strongly absorbed by biological structures, it can only effectively penetrate a few microns into typical specimens[15]. Without subsurface signals degrading image contrast, strong optical sectioning is achieved near the sample surface. This eliminates the need to prepare flat, thin samples in mobile setups. Besides, a large variety of common fluorescent dyes (e.g., DAPI, fluorescein, rhodamine, etc.) are readily excitable by sub-285 nm UV light and emit in various visible ranges, thereby providing a simple mechanism for general fluorescence microscopy. Because sub-285 nm UV is blocked by common optical materials such as borosilicate glass and plastics, it is unnecessary to filter out the excitation light with designated filters, and the entire visible range of the emitted light can be captured by an RGB camera in a single shot[15]. As a result, both the optical design and the operating procedures are simplified for a smartphone-based multichannel fluorescence microscope. Finally, sample preparation for MUSE is simple and fast. Most of the time, surface fluorescence staining can be performed with a single seconds-long soaking step followed by a brief rinse[16], a process that can be easily performed by non-professionals outside the laboratory within minutes.

The simplicity of MUSE could enable numerous microscopy applications (e.g., histology) that are otherwise difficult using mobile systems. However, there is a major engineering challenge to directly add MUSE functionality to a compact smartphone microscope. While UV illumination needs to be introduced between the sample and the microscope objective, compact smartphone microscopes often have very small working distances. The clearance between the sample and the objective is insufficient to fit most conventional sub-285 nm light-emitting diodes (LED). To address this problem, Pocket MUSE is designed to deliver light using frustrated total internal reflection (TIR)[19,20] through a UVC transparent optical window, which also serves as the sample holder that is pre-aligned at the focus of the smartphone microscope objective. By installing the 285 nm UV LED against the edge of the optical window, a significant amount of light is guided into the optical window (Supplementary Note 1.6). This not only creates a uniform illumination over the full FOV but also further simplifies the system by eliminating the need for a focusing mechanism.

In addition, while the highest resolution of current compact smartphone microscope designs is often above 3.5 μm[6,10,11], many MUSE applications (e.g., diagnostic histology) require resolution below 2 μm to effectively resolve cellular structures in biological samples. One step further, we improved the effective resolution (the resolution limited by either optical or pixel sampling) of Pocket MUSE down to the submicron level through optimization of the optical design, as described below. We also piloted a series of sample processing strategies that can be easily implemented for Pocket MUSE imaging.

## Results

**Overview of Pocket MUSE design and operation.** To ensure low cost and ease of fabrication, Pocket MUSE features a simple design while maintaining the ability to obtain high-quality images. It consists of only four major components: an objective lens, a sample holder, UV LED light sources, and a base plate (Fig. 1a). A reversed aspheric compound lens (RACL) serves as the proximal optical element, centered immediately in front of the

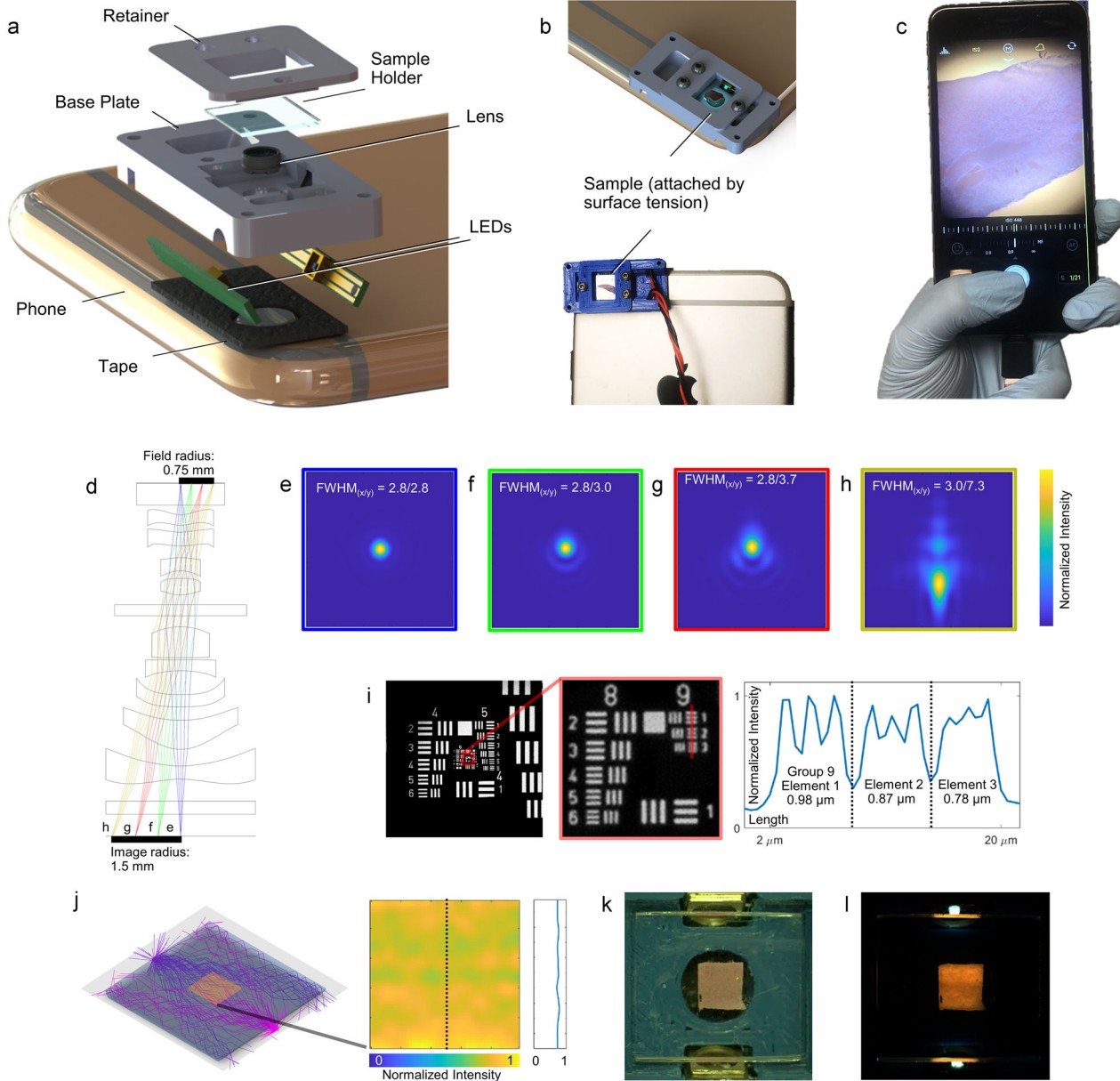

**Fig. 1 Design, simulation, and characterization of Pocket MUSE. a** Exploded schematic showing the major components of Pocket MUSE. **b** 3D rendering and photographs showing samples attached to the microscope through surface tension. **c** Photograph demonstrating the hand-held operation of Pocket MUSE. **d** Optical simulation of a small RACL (US7643225B1, f-number = 2.45) stacked on top of a smartphone camera lens (US20130021680A1, f-number = 2.46), showing ~2× magnification (1.5 mm/0.75 mm). **e–h** Simulated Huygens point spread function (PSF) at the image plane showing the full width at half maximum (FWHM) spot diameter (**e**) $\Delta y = 0$, (**f**) 0.25, (**g**) 0.5 and (**h**) 0.75 mm from the center of the field (simulation FOV: $36.42 \times 36.42$ $\mu m^2$). Minimal distortion to the PSF was observed within the center 1 mm diameter FOV. The simulation does not reflect the practical optical resolution of Pocket MUSE, which is higher due to smaller f-numbers (larger NAs). **i** Image of a USAF-1951 resolution target acquired with a 1/7" RACL (Largan 40069A1) attached to an iPhone 6s+, showing resolving power up to Group 9 Element 2 (0.87 μm) based on the Sparrow criterion. The middle image is a close-up view of the region in the red box in the original image. The plot on the right represents the normalized intensity profile at the location of the vertical red line. Additional characterization at different locations of the FOV is shown in Supplementary Fig. 4. **j** Ray tracing simulation of the dual-LED frustrated TIR setup, showing relatively uniform illumination achieved in a $2.5 \times 2.5$ $mm^2$ region at the center of the sample holder. The heat map in the middle indicates the normalized intensity of the square. The plot on the right represents the normalized intensity profile labeled with the vertical line. Intensity variation is within 10% across 3 mm FOV. **k** Bright-field macro image showing a $2.5 \times 2.5$ $mm^2$ fluorescent phantom on the Pocket MUSE sample holder. **l** A relatively uniform excitation pattern was achieved with the frustrated TIR illumination configuration, matching the simulation result.

smartphone camera, and provides a relatively wide FOV of $\sim 1.5 \times 1.5$ $\mu m^2$. The sample holder, a 0.5 mm-thick fused quartz glass optical window, has its top surface pre-aligned with the focal plane of the objective lens. This eliminates the need for fine focusing mechanics that are essential for traditional microscope designs. The required fine focusing is performed via smartphone camera focus adjustment. The sample holder also serves as a waveguide for the frustrated TIR illumination (Supplementary Note 2.2). The light sources, two miniature UV LEDs (1–5 mW, 6.5–7.5 V, 50–250 mA, 3535 packages, and 275–285 nm center wavelength), are powered directly with the smartphone battery via the USB port through a step-up regulator (e.g., Pololu

U3V12F9). All the components are integrated into an ultra-compact base plate, designed to be 3D printable using simple fused-deposition modeling (FDM), utilizing <2 g of material.

By eliminating all adjustment mechanisms, even first-time users can easily operate Pocket MUSE. To image, samples (tissue or fluid) are attached to the sample holder by surface tension (Fig. 1b). As the sample holder is pre-aligned to the focal plane of the objective (Supplementary Note 1.6), the sample is always in focus during normal operation. In addition, similar to conventional smartphone photography, Pocket MUSE is designed to take quality images while holding the phone in any orientation with one hand (Fig. 1c). This provides extra convenience for applications in the field, where a stable working bench is not always available. After imaging, the sample holder can be easily cleaned using cotton swabs and common solvents (e.g., 70% isopropanol). For heavy-duty cleaning or sterilization, the sample holder can also be detached from the device.

**Objective lens selection.** The microscope compartment of Pocket MUSE is improved over previous RACL smartphone microscope designs[10,21]. In this implementation, the RACL design delivered good resolution and a large FOV while maintaining a relatively low cost (lens cost < $10). The principle behind this design is simple and robust. Because smartphone camera lenses are capable of telecentric imaging, stacking an identical pair of such lenses face-to-face creates 1:1 finite image conjugation (object size: image size) between their back focal planes, corresponding to the object plane (sample surface) and the image plane (sensor surface) of the microscope. However, this original design had a critical limitation. While a common smartphone camera lens often has f-numbers around 1.5–2.4 (corresponding to a numerical aperture of ~0.2–0.3) and provides ~1–2-μm optical resolution, the actual resolution is pixel-limited because a typical smartphone camera sensor often has a pixel size of ~1.5–2 μm. The pixels are grouped in 2 × 2 as part of an RGB Bayer filter configuration, further reducing the effective pixel size to ~3–4 μm.

Improving the resolution of previous RACL designs would further expand the capabilities and potential applications for smartphone microscopes. The original RACL manuscript suggested that a smartphone with a large sensor and small pixel size (e.g., Nokia 808) can help improve the effective resolution[10]. However, it is not a universal solution for most smartphones. Here, we show that the effective resolution and magnification of the RACL design can be further improved using a smaller RACL with a shorter focal length (e.g., <1.5 mm). Through Zemax optical simulation (lens designs from patents by Largan Precisions[22,23]), we confirmed that a smaller RACL can effectively reduce the image conjugation ratio (e.g., from 1:1 to 1:2) (Fig. 1d) while maintaining good optical performance over a >1 mm FOV (Fig. 1e–h). While preserving optical resolution, the smaller RACL increases the magnification of the system and boosts the effective resolution through denser spatial sampling when projected onto the smartphone camera sensor. As the simulation did not take into account the specific optics and sensor size for each smartphone, we collected a wide range of lens samples obtained from aftermarket consumer products and tested them with different smartphones (Supplementary Fig. 4.3–4.8). Among the lenses we tested, we found that <1 μm effective resolution (>25% contrast on USAF-1951 test target) and 1.5 × 1.5 mm² FOV can be achieved on an iPhone 6s+ (with Sony RGBW IMX315 12MP camera sensor and F/2.2 lens with 2.65 mm focal length) using a smaller RACL designed for 1/7″ image sensors (Fig. 1i). Based on the aperture diameter (~0.9 mm) and focal length (~0.5 mm), the f-number of the lens is around 1.8 (NA

0.27). By comparison, this optical design greatly outperforms a conventional benchtop microscope with a high-quality 10× objective (Supplementary Note 4.2). Therefore, we chose to use such lenses in the Pocket MUSE design.

**Frustrated TIR illumination.** A smaller RACL often has a large entrance aperture (>3 mm diameter) and a short working distance (<1 mm). Within this narrow working distance, it is necessary to fit a sample holder (optical window). Because conventional sub 285 nm UV LEDs often have package sizes (3.5 × 3.5 × 1 mm³) that are even larger than the RACL, implementing the original MUSE illumination configuration[15,16] becomes nearly impossible due to limited spatial clearance. To overcome this problem, we identified frustrated TIR[19] (Fig. 1j) as an effective approach to deliver light to the sample surface. In our configuration, by positioning the LED die closely against the optical window, sub 285 nm UV illumination is coupled into the sample holder (a 0.5 mm thick fused quartz optical window) from the side faces of the optical window. Above the glass–air critical angle, the coupled light reflects between the two glass surfaces through TIR. When a sample is present, the glass–air interface turns into a glass–sample (glass–water) interface. It changes the TIR critical angle and allows some light to refract out of the glass, facilitating sample illumination. In addition, we further optimized our TIR illumination by implementing two LEDs. Because a significant amount of light is absorbed by sample regions closer to the LED, a single LED could not effectively illuminate the entire FOV. Through optical simulation, we noticed a >50% optical energy drop across 2 mm of the sample (Supplementary Fig. 4.9), causing significantly non-uniform illumination. To compensate for this drop, we added another LED on the opposite edge of the optical window. Through both modeling and experiments, we show that relatively uniform illumination (<±10% variation across 3 mm) can be achieved with the dual-LED setup (Fig. 1j-l, Supplementary Fig. 4.10).

**Histology imaging.** Slide-free histology is one of the most well-established MUSE applications. Therefore, as the first demonstration, we show that Pocket MUSE is fully capable of producing high-quality histology images similar to those acquired from benchtop MUSE systems[16]. To verify the performance of Pocket MUSE, we performed benchmark tests between Pocket MUSE and other imaging modalities (Fig. 2), including conventional MUSE imaging (w/ Nikon Plan APO 10×/0.45), traditional bright-field imaging using a commercial benchtop microscope (Keyence BZ-X810) with 5× (Nikon LU Plan 5×/0.15), 10× (Nikon Plan APO 10×/0.45) and 20× (Nikon Plan Fluor ELWD 20×/0.45) objectives, and bright-field microscopy with a stand-alone 1/7″ RACL on the smartphone (not the fully assembled Pocket MUSE device). Using a similar single-dip staining protocol from the original MUSE demonstration[16], we show that Pocket MUSE can produce image data and pseudo-H&E color-remapping similar to a customized benchtop MUSE microscope with a commercial objective. We also show that the RACL lens used in Pocket MUSE is able to produce H&E data similar to a benchtop microscope at 10–20× magnification.

In addition, we also demonstrated Pocket MUSE histology imaging on a large variety of tissue samples (e.g., kidney, muscle, etc.) within minutes (Fig. 3a–d). Pocket MUSE provides a similar FOV compared to a conventional 10× objective (e.g., ~1.5 × 1.5 mm² with a standard 22 mm camera sensor). With sufficient resolution to resolve individual cell nuclei, it is readily useful for a number of histology-centered applications. Using images captured from Pocket MUSE, we were also able to implement the color-remapping technique[16] to generate histology images

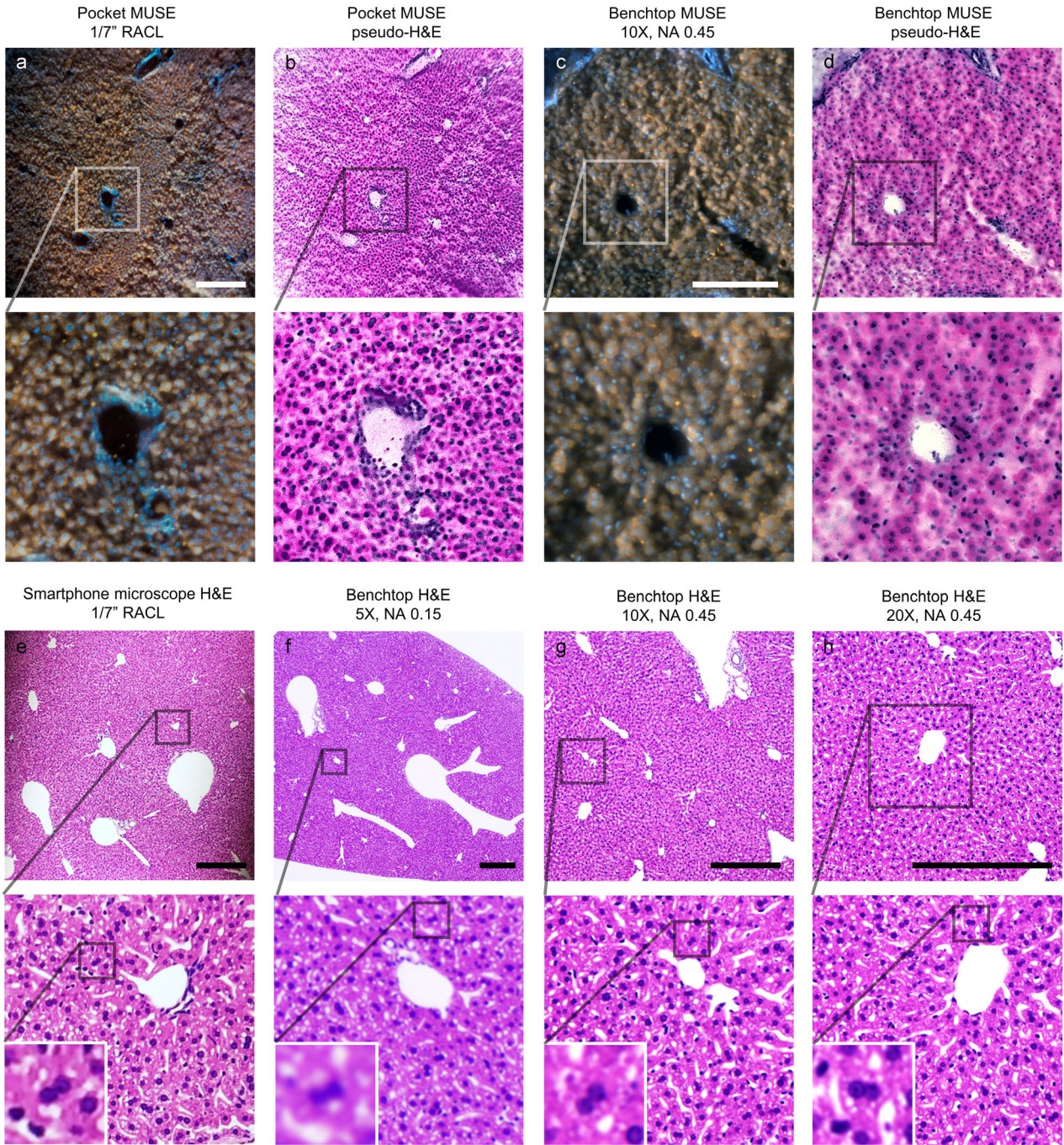

**Fig. 2 Images from Pocket MUSE and other imaging modalities. a** Pocket MUSE image of a thick mouse liver section stained with 0.05% w/v Rhodamine B and 0.01% w/v DAPI. **b** Psuedo-H&E color-remapping of (**a**). **c** Benchtop MUSE image of the same sample from (**a**). **d** Psuedo-H&E colo-remapping of (**c**). Close-up views of the region in the boxes (box size: 300 × 300 μm²) are shown below. **e** Smartphone microscopy image of a mouse liver H&E slide using the same RACL from Pocket MUSE. **f**–**h** The same H&E liver slide imaged with a benchtop microscope at 5×,10×, and 20× magnifications. 2-level close-up views of the images are shown below for (**e**)–(**h**) (Box sizes: 250 × 250 μm² for level 1 and 30 × 30 μm² for level 2). The contrast between imaging modalities is different (e.g., nucleoli have less contrast in the MUSE images). Scale bars: 300 μm.

mimicking the color contrast of conventional H&E staining (Figs. 2b, e and 3e–h).

We further demonstrate that whole-mount fluorescent immunohistochemistry (IHC) stained tissue can be imaged with Pocket MUSE (Fig. 4a). With overnight staining at a slightly higher concentration (e.g., 1% v/v compared to 0.4% v/v for conventional staining), Alexa Fluor 488-conjugated GFP antibodies provided sufficient contrast between the Thy1-GFP positive neurons and the background fluorescence. Common cell nuclei

dyes (e.g., DAPI, propidium iodide, etc.) could also be easily incorporated into the IHC staining process. Different labels can be readily separated by unmixing the RGB channels (Fig. 4b, c).

**Plants and environmental sample imaging**. Pocket MUSE is also a promising tool for imaging various plants (e.g., vegetables, algae, etc.) and environmental samples (e.g., micro-animals, synthetic pollutants, etc.). Many samples (e.g., Coriandrums, micro-plastic

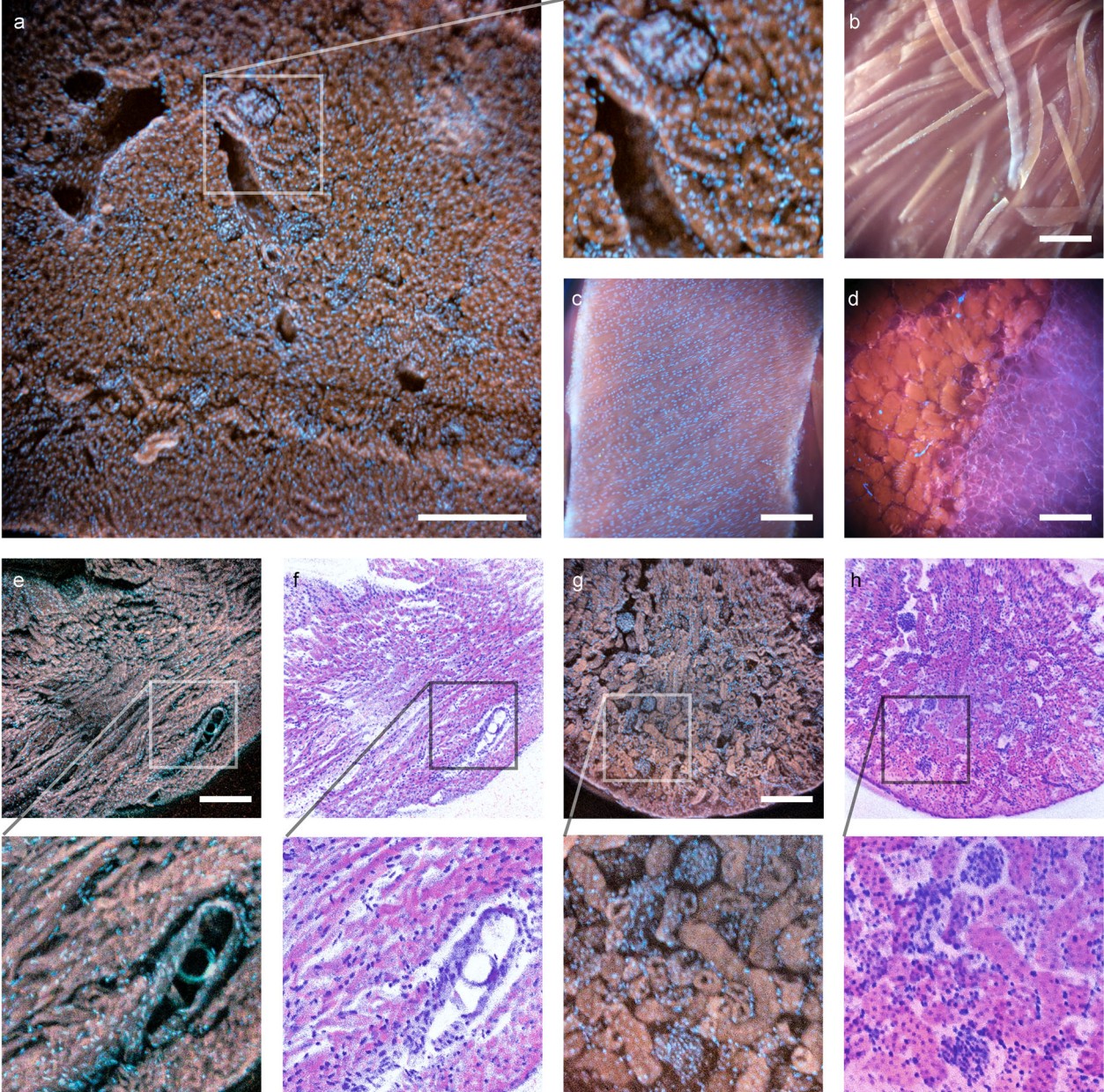

**Fig. 3 Histology images acquired with Pocket MUSE.** All samples were stained with 0.05% w/v Rhodamine B and 0.01% w/v DAPI unless otherwise specified. **a** Image of a thick section of mouse kidney sliced with a razor blade. A close-up view of the region in the white box (box size: 320 × 320 μm²) is shown on the right. **b** Image of mouse skeletal muscle torn with tweezers. **c** Image of the serosal surface of a mouse small intestine. **d** Image of salmon steak sliced with a kitchen knife. **e** Image of a thick section of mouse heart sliced with a razor blade. An additional 0.1% w/v Light Green SF dye was added to the staining solution to suppress the transmitted light. Fluorescent emission is not significantly affected. A close-up view of the region in the white box (box size: 500 × 500 μm²) is shown below. **f** Pseudo-H&E remapping of (**e**). **g** Image of a thick section of mouse kidney sliced with a razor blade. A close-up view of the region in the white box (box size: 500 × 500 μm²) is shown below. **h** Pseudo-H&E remapping of (**g**). Scale bars: 300 μm.

particles, etc.) are intrinsically fluorescent when excited around 265–285 nm. These samples are capable of generating structural contrast without any staining. In addition, both MUSE fluorescence and transmissive bright-field imaging can be achieved in a single platform. Images of a carrot taken with Pocket MUSE, conventional benchtop MUSE, bright-field, and fluorescence microscopy are shown for comparison (Fig. 5).

As with animal tissues, plant tissue could also be stained to produce additional micro-structural contrast with a single dip staining process (Fig. 6). For instance, DAPI effectively labels cell nuclei (Fig. 6a, b) and polysaccharides moieties (found in, e.g., cell walls, root saps, and starch (Fig. 6c, e–g)), while rhodamine

demonstrates accumulation in the xylem (Fig. 6c–e). We also observed that some absorptive staining (e.g., iodine-stained starch (Fig. 6g)) could be effectively incorporated with fluorescent stains to create different color contrast between different plant structures.

**Bright-field and hybrid imaging.** Pocket MUSE can also easily acquire bright-field images when UV illumination is not enabled. This provides a simple and effective method for visualizing naturally colored thin samples (e.g., blood smears) (Fig. 7a, d). A conventional fluorescence microscope requires switching the filter cube to an open setting for bright-field microscopy which is

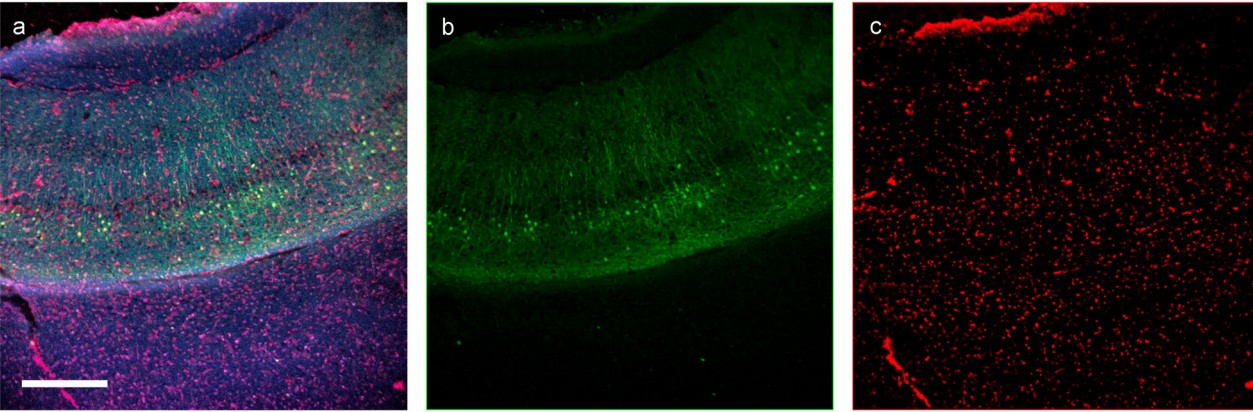

**Fig. 4 IHC image of a 500 µm thick Thy1-GFP brain slice acquired with Pocket MUSE.** The sample was stained with anti-GFP antibody (Alexa Fluor 488 conjugate) and propidium iodide. **a** RGB image acquired with Pocket MUSE. **b** Alexa Fluor 488 signal (from thy1-GFP) unmixed from the green channel, showing Thy1 positive neurons. **c** Propidium iodide signal unmixed from the red channel of the RGB image, showing cell nuclei. Scale bar: 300 µm.

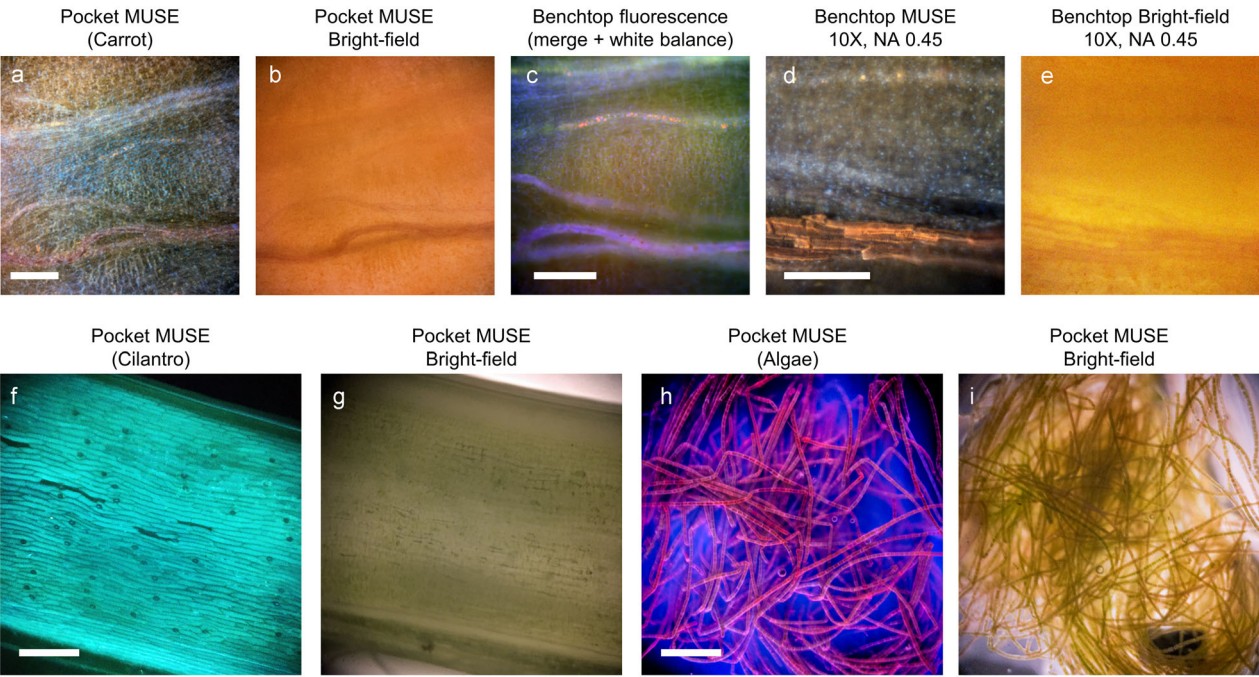

**Fig. 5 Comparisons between Pocket MUSE imaging and other imaging modalities.** A slice of carrot stained with 0.05% w/v rhodamine B and 0.01% w/v DAPI was imaged with (**a**) UV illumination on Pocket MUSE. **b** bright-field illumination on Pocket MUSE, (**c**) a benchtop fluorescence microscope using 405/488/594 nm excitation light and a 10× objective, (**d**) benchtop MUSE microscope, and (**e**) benchtop microscope with bright-field illumination. Additional comparisons of Pocket MUSE images with MUSE illumination (left) and bright-field illumination (right) show (**f**) and (**g**) the surface of a cilantro stem (no staining), and (**h**) and (**i**) a cluster of filamentous algae stained with 0.05% w/v propidium iodide followed by rinsing in 1% w/v hydroquinone. Additional images are shown in Supplementary Fig. 4.11. Scale bars: 300 µm.

difficult in a compact smartphone microscope. Because Pocket MUSE does not rely on filters, no mechanical switching is required to change between fluorescence and bright-field imaging. Trans-illumination bright-field microscopy can be realized simply by directing the sample holder towards a bright diffusive surface (e.g., white wall, printing paper, etc.) in the far-field. Regular room light and/or natural light (>100 lumens/m$^2$) provide sufficient illumination.

Overlaying the fluorescence and bright-field images is a common and useful technique to highlight the structures of interest in biological samples. With Pocket MUSE, fluorescence and bright-field contrasts can be combined through a single capture simply by enabling the UV illumination during bright-field imaging (hybrid mode) (Fig. 7c, e). As an example, with a thin blood smear, we demonstrate that white blood cells (WBC, fluorescence) can be highlighted in a crowd of red blood cells (RBC, bright-field) by simply mixing in a small amount of fluorescent nuclei dyes (e.g., 0.01% w/v acridine orange) in the specimen (Fig. 7e).

**Mucosal smear imaging**. Mucosal smears are used in many medical diagnostic applications, such as Pap smears. Mucosal smear preparation for Pocket MUSE is extremely simple and can be performed within 30 s. The specimen is collected with a cotton swab that is then dipped in a dye (e.g., propidium iodide with CytoStain™), briefly washed in tap water, and smeared onto the sample holder (Supplementary Video 1). Compared to bright-

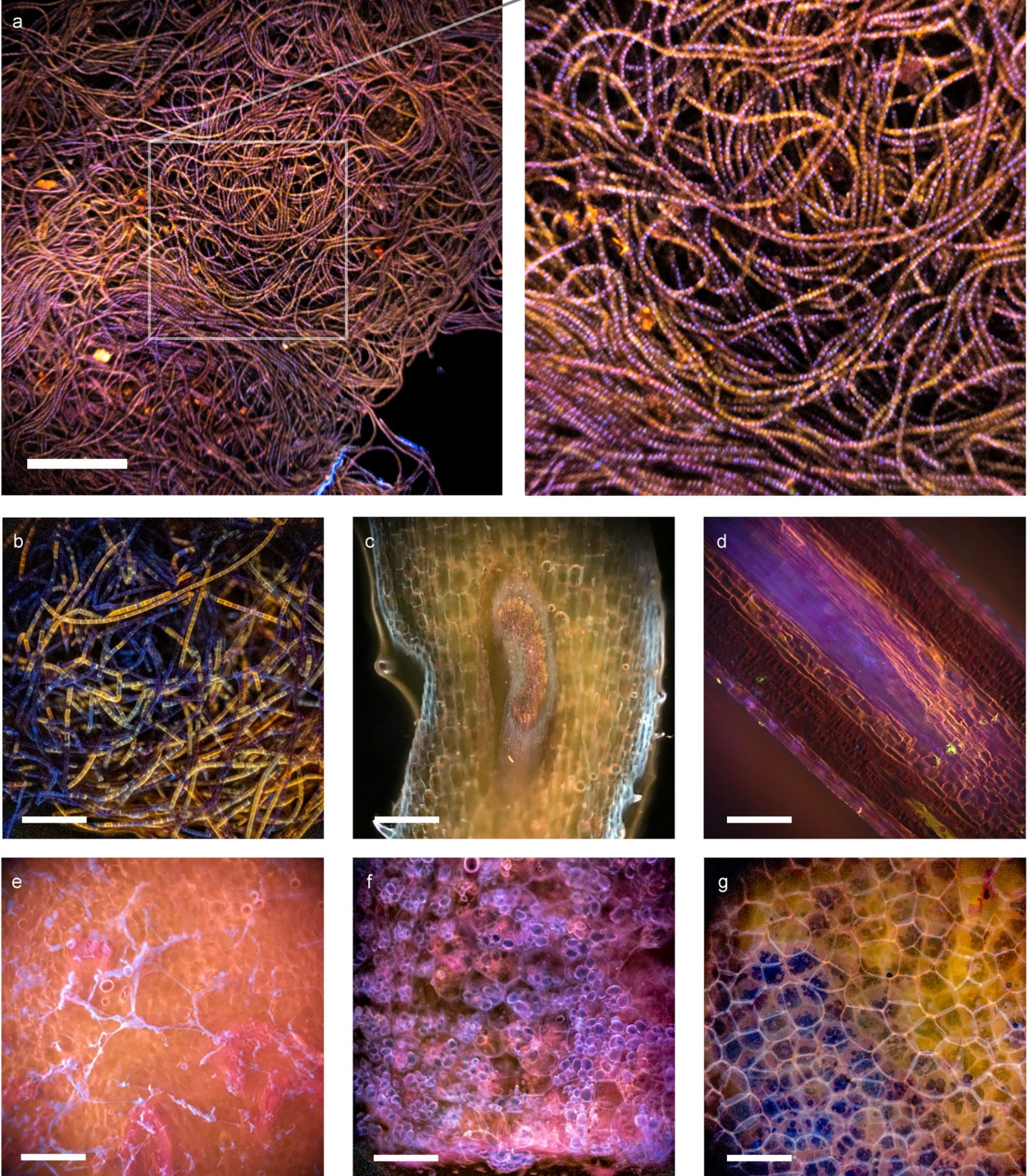

**Fig. 6 Images of plant samples acquired with Pocket MUSE. a** A cluster of filamentous algae stained with 0.05% w/v rhodamine B and 0.01% w/v DAPI. A close-up view of the region labeled with the box (box size: 600 × 600 μm$^2$) is shown on the right. **b** A cluster of different filamentous algae, (**c**) a cross-section of a clover stem, (**d**) a cross-section of a pine needle, (**e**) a cross-section of onion, and (**f**), (**g**) cross-sections of potato stained with the same rhodamine B and DAPI solution. The potato slice in (**g**) is further washed in saturated iodine water. The stain effectively created a large variety of structural contrast in different plant samples. In general, rhodamine B has a high affinity to xylem structures (**c**) and (**e**), while DAPI stains carbohydrate abundant structures such as cell walls (**c**), and (**f**), root saps (**e**), and starch granules (**f**). In the filamentous algae sample (**b**), rhodamine B appears to have a higher affinity to some algae cells (appears yellow), while some cells are only stained with DAPI (appears blue). Starch granules are stained in black by elemental iodine, indicating absorptive stains may work collaboratively with fluorescent stains under MUSE contrast. Scale bars: 300 μm.

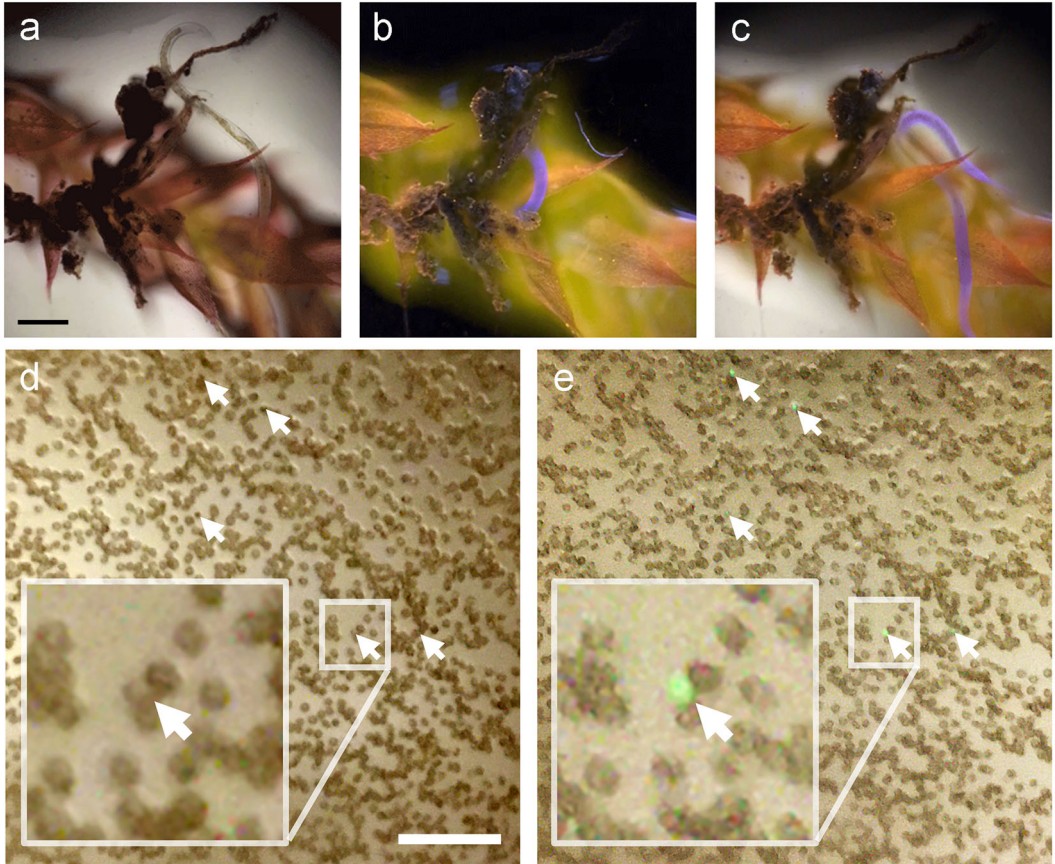

**Fig. 7 Examples of bright-field and hybrid mode imaging. a–c** A live roundworm (the purple-blue structure in (**b**) and (**c**)) moves around a piece of grimmia moss (Supplementary Video. 3). The sample was stained in 0.05% w/v rhodamine B and 0.01% w/v DAPI, washed in tap water, and imaged with (**a**) bright-field mode, (**b**) MUSE fluorescent mode, and (**c**) hybrid mode (simultaneous bright-field and fluorescence). **d, e** A 5% v/v blood sample stained with ~0.01% w/v acridine orange. The sample was smeared on the sample holder surface, and the same region was imaged with (**d**) bright-field mode and (**e**) hybrid mode. Most disk-shaped RBCs are distinguishable under the bright-field contrast. White arrows point to the cell nuclei of potential WBCs stained with acridine orange which does not show up in the bright-field image. A region of each image, labeled with the smaller box (box size ~70 × 70 μm$^2$), is zoomed four times and shown in the bigger box. Identifying RBCs and structures with nuclei in a blood smear has diagnostic potential. Scale bars: 100 μm.

field imaging on the Pocket MUSE, the cytology staining results in a significantly higher contrast between the cell bodies, nuclei, and the background under MUSE fluorescent contrast. Cell morphology can be visualized over the majority of the FOV due to low aberrations at the edges (Fig. 8). In addition, as conventional mucosal smear cytology imaging requires cells to be attached to a flat glass surface, Pocket MUSE allows cells to be imaged directly on the cotton fiber matrices. Because MUSE captures the surface, some volumetric aspects of cell morphology can be visualized. Finally, although only a single FOV could be imaged, a larger population of cells could be rapidly reviewed by the repeated repositioning of the same swab.

**Selective bacteria imaging**. Fluorescent staining has been widely used to examine bacteria in liquid samples[24–27]. As a preliminary demonstration, a bacterial suspension was labeled with fluorescent dyes (e.g., acridine orange) in a simple mixing step. Individual bacteria are smaller than the resolution limit of Pocket MUSE, but their presence can be effectively visualized if sparsely dispersed in a fluid sample. Suspended bacteria show a distinct twinkling in preview mode due to their movement in and out of the focal plane (Supplementary Video 2). In addition, with bacteria-specific fluorescent probes[24,25,28], Pocket MUSE could also differentiate different populations of microorganisms. As a

preliminary demonstration, we show that nucleic acid stain (DAPI, which labels all bacteria) combined with peptidoglycan staining (wheat germ agglutinin Alexa Fluor 594 conjugate (WGA-AF594), which labels gram-positive bacteria) can differentiate *Bacillus subtilis* (Gram-positive) and *Escherichia coli* (Gram-negative) bacteria populations based on the color of microbe particles (Fig. 9a–h). By analyzing the bright speckles in the data one can visualize the distribution of pixel values (2D bivariate histogram, Fig. 9i–l) and determine the proportion of different populations of bacteria in the sample.

## Discussion
Pocket MUSE is a simple and effective solution for many microscopy applications. Without sacrificing the cost advantage of a compact smartphone microscope design, we identified a high-performance single-lens solution that can produce high-quality images that are comparable to conventional benchtop microscopes at low magnifications (e.g., a 10× objective, Supplementary Fig. 4.2). As the resolution of previous RACLs is mostly limited by the pixel resolution of the smartphone camera, using RACLs with shorter focal lengths allows the pixel resolution to approach the diffraction-limited optical resolution, resulting in improved imaging performance. In addition, deploying the MUSE approach adds fluorescent imaging functionality while

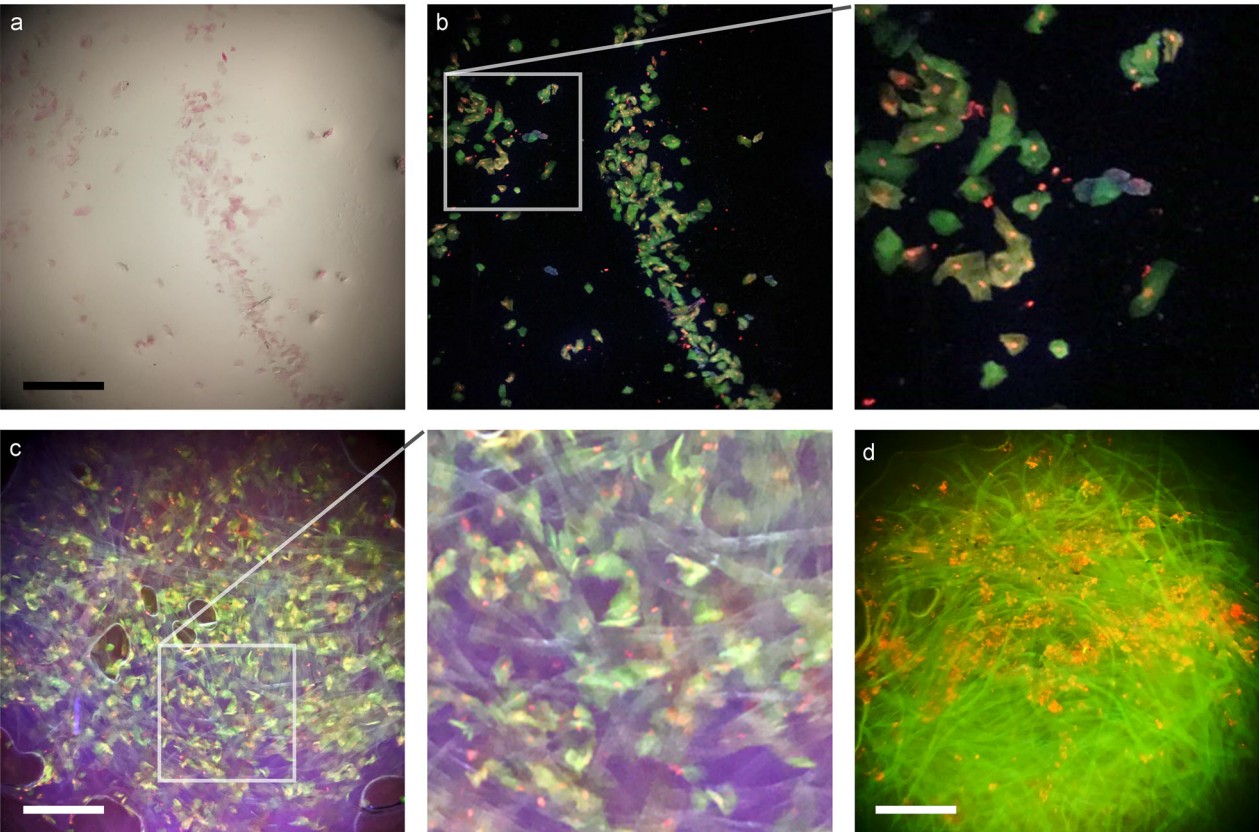

**Fig. 8 Images of mucosal smear samples acquired with Pocket MUSE.** A cheek swab sample was acquired by smearing the cells on the Pocket MUSE sample holder and the same region was imaged with (**a**) bright-field mode and (**b**) MUSE fluorescent mode. The sample was collected using a conventional cotton swab, stained in 10% v/v CytoStain with 0.05% w/v propidium iodide, and washed with tap water. A close-up view of the region labeled with the box (box size: 500 × 500 μm$^2$) is shown on the right demonstrating good image quality even near the corner of the FOV. The cytoplasm is stained in green or yellow by CytoStain. Cell nuclei (within the cytoplasm) and bacteria (outside cytoplasm) are stained in red by propidium iodide. **c** MUSE fluorescent images of a cotton swab tip containing a stained cheek swab sample. A close-up view of the region labeled with the box (box size: 500 × 500 μm$^2$) is shown on the right showing cells attached to the cotton matrix without significant flattening from the glass surface due to surface tension. **d** MUSE fluorescent images of a cotton swab tip containing a healthy conjunctival (eyelid) swab sample stained with CytoStain and propidium iodide. The samples showed fewer cells, but more nucleic acid structures which are likely from conjunctival microorganisms. The cellulose matrices were rapidly stained in green due to the lack of viscous mucus. Scale bars: 300 μm.

also simplifying the sample preparation and operating procedures. Through a number of preliminary demonstrations, we showed that Pocket MUSE could be used to image a wide variety of samples prepared with simple procedures. With minimal or no modifications, the demonstrated techniques are readily applicable to many real-world microscopy applications.

With relatively low cost, Pocket MUSE is also simple to fabricate and easy to operate. For small volume (e.g., <5 pieces) production, all of the parts are available from major online vendors and the material cost of the optical add-on to a smartphone is in the range of $20–50 depending on the regional retail prices of the parts (Supplementary Note 1.8). Of course, these prices could be markedly lower if the units were produced in volume. The fabrication process of Pocket MUSE is optimized for entry-level researchers and engineers, only requiring basic prototyping knowledge and tools that are widely available (Supplementary Note 1.9). As the design involves no adjustment mechanisms (e.g., focusing) and operating the system is almost as easy as taking smartphone photos, no significant training is required even for non-professional users.

It is a challenge to identify the appropriate RACL for Pocket MUSE because most aspheric compound lenses are only available as original equipment manufacturer (OEM) components for consumer electronics. While it is difficult to purchase these lenses

directly from the manufacturers in small quantities, some lenses are easily found in various common consumer electronics, especially in aftermarket parts (e.g., replacement cameras). Although we have only identified two ideal lens modules (Largan 40069A1 and Lenovo ThinkPad X240 webcam lens), any lens with similar technical specifications (e.g., f-number, focal length, etc.) should function similarly in our design. As shown in the result section, Additional information about lens selection is discussed in the supplementary material (Supplementary Note 1.1).

Frustrated TIR illumination implemented in Pocket MUSE may be confused with the similar-sounding TIR fluorescence (TIRF) microscopy[29–31]. In conventional TIRF microscopy, fluorophores are excited by the evanescent field at the sample-glass interface, resulting in thin optical sectioning. This configuration often requires well-collimated illumination and fine angular alignment to prevent refractive leakage at the glass–sample interface, requiring expertize in optics. However, in Pocket MUSE, 285 nm UV readily provides good optical sectioning capability. Refraction-based frustration is desired for extended depth of field when imaging samples with uneven surfaces and/or dispersed in a solution. Therefore, the UV LED can be positioned next to the optical window without fine angular alignment (Supplementary Note 1.5–1.6). Besides, as demonstrated by Yoshitake et al.[17], optical sectioning of MUSE imaging

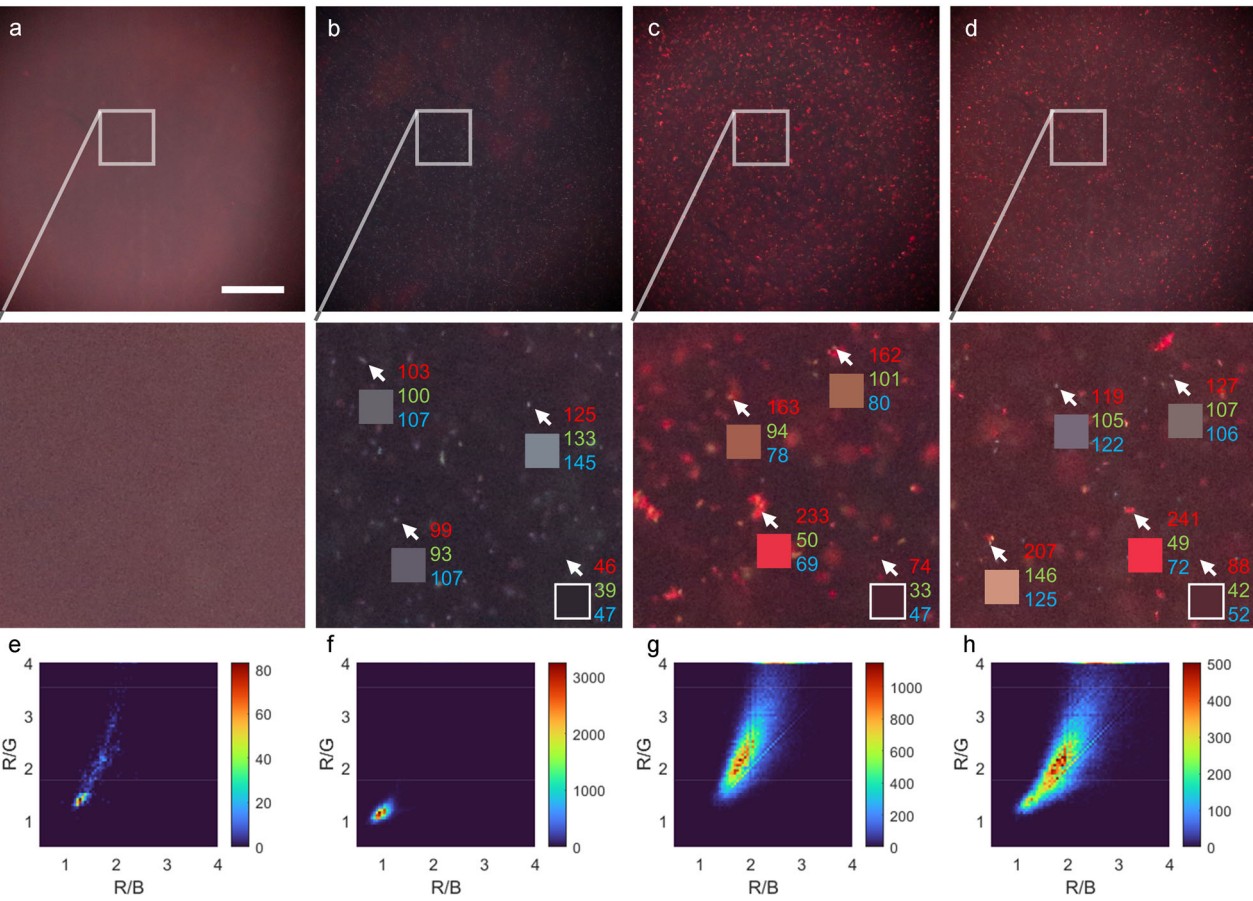

**Fig. 9 Images of bacterial samples acquired with Pocket MUSE.** Two different populations (Gram + and Gram-) of bacteria can be separated with Pocket MUSE. Aliquots of (**a**) deionized water, (**b**) an *E. coli* (Gram-) culture, (**c**) a *B. subtilis* (Gram + ) culture, and (**d**) a mixture of E. coli culture and *B. subtilis* culture imaged with Pocket MUSE. All samples were stained with 0.01% w/v DAPI and 0.02 % w/v WGA-AF594, and imaged under the same conditions. Images show (**a**) no bright spots, (**b**) blue-grayish spots, (**c**) a mixture of reddish and orangish spots, and (**d**) a mixture of blue-grayish, reddish and orangish spots dispersed across the FOV. For each image, a close-up view of the region labeled with the box (box size: 250 × 250 μm²) is shown below. For (**b**)–(**d**), the center pixels of three bright spots in each image, pointed to by the arrows, are shown in the boxes below. The RGB values (top to bottom) of the pixels are labeled on the side. The nucleic acid in *E. coli* (DAPI stain) contributes to the blue-grayish speckles (R ≈ G ≈ B). Peptidoglycan on the *B. subtilis* surface (WGA-AF594 stain) contributes to the reddish speckles (R > G + B). Nucleic-acid-rich endospores in *B. subtilis* contribute to the orangish speckles (R > G, R > B & R < G + B). **e**–**h** bivariate histogram representation of the bright pixels in (**a**)–(**d**), showing the distribution of colors. All images were median-filtered and background-corrected before analysis. Only bright pixels are counted (e.g., four times greater than the mean value of the corresponding image). X-axes represent the ratio between red and blue channels. Y-axes represent the ratio between red and green channels. Scale bars: 300 μm.

is improved by water immersion illumination. As frustrated TIR restricts the possible ranges of incidence angles at the glass–sample interface, better optical sectioning can be achieved compared to conventional air immersion MUSE imaging (Supplementary Note 2.7). In addition, since the LED is the most expensive component of Pocket MUSE, it is still worth considering the use of a single LED when the cost is a major concern. Non-uniform illumination with a single LED may not be a significant problem for many applications and can be corrected with straightforward image post-processing steps.

Pocket MUSE generated histology data of a similar quality to conventional benchtop MUSE systems[16] (Figs. 2 and 3). Pseudo-H&E color-remapping was also reproduced to help better elucidate the Pocket MUSE-generated histology images with more histology familiar color contrast. Although higher magnification and image stitching functionalities are limited with the current design, it is readily useful for quick evaluation of small histology samples such as skin biopsies immediately after sample extraction. In addition to dye-based structural imaging, we also demonstrated whole-mount fluorescent IHC with Pocket MUSE

(Fig. 4). Although IHC staining was proposed as a potential application in previous MUSE demonstrations, it has only been demonstrated with quantum dots conjugated to antibodies[32]. To our knowledge, this is the first demonstration of MUSE IHC imaging using fluorophore-conjugated antibodies. Considering our demonstrated IHC performance, Pocket MUSE could be a promising research and teaching tool when access to fluorescence microscopes and microtomes is limited.

Pocket MUSE produced high-quality images of plant and environmental samples, which could be extremely useful for STEM education[33] and various research in the field (Figs. 5 and 6). For many types of samples (e.g., algae, root vegetables, etc.), 275–285 nm UV-induced fluorescent contrast is also more informative and visually pleasing compared to conventional bright-field contrast. Considering Pocket MUSE is also more affordable, durable, portable, and user-friendly (e.g., avoiding thin sectioning) compared to conventional benchtop microscopes, it offers additional benefits for STEM teaching both in and outside the classroom. In addition, Pocket MUSE is also a multifunctional microscope for various non-educational applications on-site. By

generating results in real-time, it is extremely beneficial for time-sensitive tasks such as monitoring water quality (e.g., algae population) and plant disease control.

Pocket MUSE could provide a promising and affordable solution to much point-of-care health monitoring and global health challenges. For instance, fluorescence microscopy has been implemented as a tool for parasite detection in resource-limited settings. In blood samples, various protozoan parasites (e.g., malaria, trypanosome, etc.) can be effectively labeled with fluorescent nuclei dyes such as acridine orange in a single step[34,35]. By introducing fluorescence functionality in a simple and cost-effective smartphone-based system, Pocket MUSE shows great potential for these applications in resource-limited settings. Image of the blood sample acquired through the hybrid mode (Fig. 7) provides a promising proof of concept for detecting sparse structures with high nucleic acid abundance. Bright-field information further confirms the location of each signal within the sample (e.g., inside or outside an RBC). Other body fluid and cytology samples, such as urine (for hematuria), semen (for sperm activity), and vaginal discharge (for fungus infection), could also be examined with Pocket MUSE through similar strategies.

Pocket MUSE enables selective observation of microorganisms in fluids samples. Bacteria are common pollutants in food and water, resulting in a large number of fecal-oral transmitted diseases. Currently, many strategies exist for characterizing bacteria in fluid samples, including direct visualization through fluorescence microscopy[24,25,28]. With simple differential fluorescent labeling, we showed that Pocket MUSE could effectively distinguish different populations of bacteria in a water sample (Fig. 9). Many existing fluorescent labeling approaches[36,37] should be readily compatible with Pocket MUSE for more specific and sensitive bacteria detection.

Pocket MUSE is more than a low-cost fluorescence smartphone microscope. It can be readily integrated into mobile sensing applications requiring multiplexed fluorescent-colorimetric detection in small areas. Many emerging diagnostic technologies, such as paper-based microfluidic devices, fluorescent immuno sensors, microarrays, and lateral flow assays[38–41], use fluorescence imaging-based detection. Pocket MUSE could potentially be a highly synergistic readout platform for these technologies, further reducing the cost, minimizing the size, improving the efficacy, and bringing them to the point of care.

It is worth mentioning that extensive UV light exposure can be unsafe. However, the UV LEDs used in Pocket MUSE are low power and widely used in various consumer devices (e.g., sanitizing systems). Also, most of the UV light is contained within the waveguide and absorbed by the sample. When properly assembled, UV leakage from the device is extremely limited. Currently, there are no regulatory concerns for the small amount of UV exposure ($<0.01$ mW/cm$^2$ in 10 cm distance) during the regular operation of Pocket MUSE. Still, it is well-known that extensive UV exposure causes damage to the skin and eyes. Therefore, proper personal protection equipment (e.g., gloves and eyeglasses) is recommended.

In summary, Pocket MUSE is not only a promising tool for various research and medical applications but also a highly accessible microscopy platform for users at all skill levels. With this extremely simple and robust design, high-quality microscopy could be performed on consumer mobile devices at a low cost. In addition, MUSE functionality further simplifies the sample preparation processes for a number of different types of samples, including animal tissue, plant samples, cytology smear samples, and micro-organisms. It shows promising potential for a wide range of mobile microscopy applications for research, diagnostics, art, and STEM education, and could be a useful tool for many microscopy tasks, especially in resource-limited settings, point-of-care diagnostics, and even consumer health monitoring applications.

## Materials and methods

**Fabrication**. Aspheric compound lenses, UV LEDs, fused silica/quartz optical windows, and other general supplies were purchased from various online vendors (Supplementary Note 1.8) and modified as follows: (1) aspheric compound lenses were gently removed from the aftermarket replacement cameras using plastic tweezers; (2) fused quartz windows of the LEDs were removed using a razor blade and the height of the LED packaging was further reduced to ~1 mm (from ~1.25 mm) by manual sanding with a file (180 Grit); (3) fused quartz optical windows were cut into ~10 × 10 mm$^2$ squares using a diamond scribe, with two opposite edges polished sequentially using 40/30/12/9/3/1/0.3 µm grade lapping films. The base plate and the sample holder retainer were designed with Solidworks, and 3D printed with polylactide using an FDM printer (Snapmaker). The modified LEDs were soldered on customized printed circuit board (PCB) adaptors (designed with Autodesk EAGLE/fabricated by OSHPark.com). The LEDs were wired to a DC up-regulator with a push-button switch in between. The components were assembled as shown in Fig. 1a, and more detailed information about the fabrication process is discussed in the supplementary materials (Supplementary Note 1 and 1.1–1.9, Supplementary Fig. 1.1–1.10, and Supplementary Table 1.1 and 1.2). Some advanced design considerations are also discussed in the supplementary materials (Supplementary Note 2 and 2.1–2.7 and Supplementary Fig. 2.1–2.8).

**Alignment**. We developed an easy and robust alignment procedure to tolerate the limited accuracy of inexpensive components (e.g., 3D printing and optical window thickness) and allow nonprofessionals to align the system. It is critical to align the sample holder to the focal plane of the RACL. To tolerate variations from the manufacturing process, the base plate is designed to be slightly thicker, so the focal plane of the RACL offsets ~150 µm below the sample surface (Supplementary Note 1.5–1.6). Alignment of Pocket MUSE is an iterative process where the baseplate surface facing the smartphone is sanded with 1000–3000 grit sandpaper until the sample surface is in focus. Taking advantage of the focus adjustment function of smartphone cameras, the focal plane of the microscope can swing by tens of microns, reducing the accuracy needed from the sanding step. The thickness of the base plate (measured with a caliper) and alignment of the system (evaluated qualitatively by image sharpness) are verified regularly (e.g., every ~30 µm) until good alignment is achieved.

**Pocket MUSE imaging**. Most Pocket MUSE images in this manuscript are acquired with the same Pocket MUSE device and an iPhone 6s+ smartphone unless otherwise specified. The Pocket MUSE component is mounted onto the smartphone with double-sided tape. The DC up-regulator is either connected to the smartphone USB outlet (for Android phones), Lightning outlet (for iPhones, with an On-The-Go (OTG) converter), or an external battery. Most samples can be mounted on the fused quartz optical window with surface tension. For samples that are thin and soft (e.g., fresh animal tissues), surface tension can effectively flatten the sample surface. For rigid samples (e.g., plant roots), it is easy to create a flat imaging surface with razor cutting. For samples with strong surface irregularities (e.g., fixed muscular tissue), additional mechanical support (e.g., 3D printed holder) could be helpful to hold the sample against the optical window, if flat imaging surfaces cannot be created with surface tension. Additional considerations for sample loading are described in Supplementary Note 3.2 and Supplementary Fig. 3.2.

After samples are loaded on the sample holder, microscopy images can be taken directly with the default smartphone camera apps. For advanced controls of imaging parameters (e.g., ISO (gain), exposure time, focus, output format, etc.), it is helpful to use third-party or customized camera apps (e.g., Halide). For MUSE imaging, UV illumination should be enabled with the push button switch before the focus and exposure adjustments. Exposure time varies between 10 ms and 1 s depending on the sample type and dye concentration. Smaller ISO (gain) is desired for a better signal-to-noise ratio. In most cases, hundreds of milliseconds exposure is sufficient for ISO 400 in the iPhone 6s (a detailed list of ISO/exposure of most main figures is provided in Supplementary Table 4.1 for reference). To prevent background light, external lights can be dimmed or aluminum foil can be used to cover the microscope. Bright-field transillumination is achieved by facing the smartphone towards a white scattering surface (e.g., white wall, printing paper, etc.). Instability by hand is usually well tolerated because relative sample motion with respect to the smartphone is extremely small, especially for exposure <250 ms. In-depth instructions about using Pocket MUSE are provided in Supplementary Note 3.

**Data processing**. Unlike scientific cameras, smartphone camera apps usually automatically process raw image data and save the data as 24-bit RGB color images. Therefore, data processing (e.g., white balance, digital filters, etc.) can take place even before (e.g., in preview mode) an image is acquired. Although it is difficult to determine the actual data processing algorithm performed by different

smartphones, such information is not required for most Pocket MUSE applications. Still, it is possible to use third-party camera apps (e.g., Halide on iOS and ProCam on Android) to save raw (unprocessed) image data, which is especially beneficial when extended dynamic range, lossless data, and advanced processing are needed. To visualize camera raw data, it is necessary to first convert the data (e.g., DNG file) into 24-bit RGB formats (e.g., TIFF). Data conversion can be performed with software such as Adobe Camera Raw (in Photoshop) and Raw-Therapee (in GIMP). These programs are commonly used for non-scientific photo editing, so they could be easily adapted by non-professional users. Most images demonstrated in this manuscript were acquired in raw format and converted into TIFF with Camera Raw. Additional guidelines about data acquisition and processing are described in the supplementary material (Supplementary Note 3.3 and 3.4, Supplementary Fig. 3.3–3.6).

**Whole-mount samples.** Excised mouse tissue was obtained from unrelated studies with IACUC approval. The tissue was either used fresh right after dissection or fixed in 4% paraformaldehyde overnight and stored in 1X phosphate-buffered saline (PBS) at 4 °C. Other animals and plant samples were collected from the author's kitchen (e.g., vegetables, meat, etc.), university campus (e.g., algae, pine needles, etc.), and backyard (e.g., garden plants, roundworms, etc.). All samples were manually cut or torn with tweezers into smaller pieces ($<3 \times 6 \times 3$ mm$^3$). For each sample, at least one relatively flat imaging surface is created. Staining solutions were prepared by dissolving dyes in 30–70% v/v alcohol. One commonly used staining solution in this study is 0.05% w/v Rhodamine B and 0.01% w/v DAPI in 50% v/v methanol, which was used for most histology samples and some plant samples. Information about other dyes and staining solution formulation is described in the supplementary material (Supplementary Note 3.1, Supplementary Fig. 3.1). The sample is immersed in the staining solution for 5–20 s, rinsed with tap water, and briefly dried with an absorbent material (e.g., tissue paper). Pseudo H&E color remapping was performed using the method described previously[16].

For the IHC staining demonstration, a slice of fixed Thy1-GFP (Jackson Laboratory, CAT# 011070) mouse brain (500-μm thick) was obtained from unrelated studies with IACUC approval. A universal buffer (e.g., for blocking, staining, and washing) containing 3% v/w bovine serum albumin, 1% v/w Triton X-100, 0.05% v/w sodium azide, and 1X PBS was prepared ahead of time. For blocking, the brain slice was first incubated in an excess amount of the universal buffer for ~2 h at 37 °C. For whole-mount staining, the blocking buffer was then replaced with 500 μL fresh universal buffer containing 1% v/v GFP Polyclonal Antibody (Alexa Fluor 488 conjugate, Thermo Fisher Scientific, CAT#A-21311) and 0.05% w/v propidium iodide. The sample was shaken at 37 °C for 16 h. After staining, the sample was washed again in an excess amount of the universal buffer for ~2 h at 37 °C, followed by a 30 min wash in PBS. Channel unmixing was performed using ImageJ/FIJI[42].

**Cytology samples.** Blood samples were collected from one author of this paper with a consumer lancing device (for blood glucose monitoring). The experiment was determined as a non-human subject research project by the university's Institutional Review Board (IRB) and was conducted with the consent of the author who provided the sample. 100 μL of blood was mixed in 100 μL PBS containing 4 mM ethylenediaminetetraacetic acid and 0.01% w/v sodium azide. For nuclei staining, 10 μL of the blood sample was mixed with 1 μL of 0.1% w/v acridine orange in 50% v/v methanol. For dense blood smear imaging, 1 μL of the stained sample was dropped on the sample holder and air-dried prior to imaging. For thin blood smear imaging, the stained sample was further diluted 10 times with PBS prior to imaging. Similarly, cheek swab samples were collected from one author of this paper using consumer cotton swabs (Supplementary Video 1). The experiment was also determined as a non-human subject research project by the IRB and was conducted with the consent of the author who provided the sample. After swabbing the inner surface of the cheek, the cotton swab was dipped in a staining solution containing 10% v/v CytoStain (Richard-Allan Scientific) and 0.01% w/v propidium iodide for 5 s. The cotton swab was then briefly rinsed with tap water and dried with absorbent material. The stained cheek cells were either imaged after being smeared on the sample holder surface or directly on the cotton swab.

**Bacteria samples.** To test non-specific bacterial labeling, a random mixture of bacteria was collected from the cloudy supernatant of a mouse tissue specimen that was improperly stored in non-sterile PBS at 4 °C for 6 months. The sample was diluted ten times with PBS, and 100 μL of the sample was mixed with 10 μL of 0.1% w/v acridine orange in 50% v/v methanol. 2.5 μL of the mixture was dispensed on the Pocket MUSE sample holder and the aliquot was imaged directly with Pocket MUSE. To test Gram-specific bacterial labeling, Escherichia coli (E. coli) was generously provided by James Seckler from an unrelated study. Bacillus subtilis (Ehrenberg) Cohn (ATCC, CAT#23857) was ordered from American Type Culture Collection. Both bacteria were cultured in lysogeny broth overnight at room temperature. For the experiment, four samples were prepared as follows: (1) 500 μL PBS as a control; (2) 100 μL E. coli culture in 400 μL PBS; (3) 100 μL B. subtilis culture in 400 μL PBS; (4) 50 μL E. coli culture and 50 μL B. subtilis culture in 400 μL PBS. Each sample was mixed with a 100 μL staining solution, containing 0.05% w/v DAPI and 0.1% w/v WGA-AF594 in 50% methanol. 2.5 μL of each mixture

was imaged with Pocket MUSE with the same camera configuration. Matlab was used to create the 2D bivariate histogram plots (Fig. 9e–h). In brief, the images were processed with a $3 \times 3$ median filter, followed by background subtraction with $50 \times 50$ kernels. Pixels with value $((R + G + B)/3)$ greater than four times the mean value were selected and used in the bivariate histograms. Ratios between Red/Green channels and Red/Blue channels are calculated and plotted in Fig. 9e–h.

**Additional imaging experiments.** An H&E slide of fixed mouse liver was prepared by a professional clinical histology lab (University Hospitals, Cleveland, OH). Benchtop MUSE imaging was performed following the procedure described in the previous publication using the same sample from the Pocket MUSE experiments. In brief, a customized benchtop inverted MUSE system was constructed with a Nikon Plan APO 10×/0.45 objective (Nikon MRD00105), an InFocus dynamic tube lens (Edmund Optics #33-137), a Blackfly S colored CMOS camera (Edmund Optics #11-516), a 45 mW 285 nm UV LED (Thorlabs #M285L4), and conventional optical supplies and optomechanics (e.g., UV fused silica convex lenses, lens tubes, translational stages, posts, etc.). The sample was placed on a fused quartz microscope slide during imaging. Conventional bright-field and fluorescence imaging was performed using a Keyence BZ-X800 benchtop microscope (sensor format 1920 × 1440). Smartphone bright-field microscopy of the H&E slide was performed by attaching a 1/7″ RACL on a smartphone (e.g., the Pocket MUSE setup without the sample holder and the UV LED). The slide was placed on a Z translational stage with a diffused white light source installed under the slide. After focusing, the image was taken using the default camera app on the smartphone.

**Reporting summary.** Further information on research design is available in the Nature Research Reporting Summary linked to this article.

## Data availability
Additional data from this study are available in supplementary materials and from the corresponding author upon request.

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

## Acknowledgements

This work was partially supported by the National Institute of Health under awards R01EB028635. We would like to thank Junqi Zhuo for providing algae samples (from his aquarium tank) and James Seckler for providing bacterial samples. We would also like to thank Junwoo Suh for taking Supplementary Video 1.

## Author contributions

Y.L. designed the system, fabricated the devices, conducted the experiments, and wrote the manuscript. M.W.J. supervised the project and provided essential guidance through the process. A.M.R. provided guidance on potential applications of the system. R.M.L. and F.F. provided guidance on MUSE microscopy setup and MUSE sample preparation. All authors discussed the results and edited the manuscript.

## Competing interests

The authors declare the following competing interests: Y.L. and M.J. filed provisional patent applications related to this manuscript. R.M.L. is listed as the inventor of two Patents related to the MUSE technology described in this manuscript. R.M.L. and F.F. are co-founders of HistoliX, Inc.
