## [Peer Review File · Communications Biology]

Reviewers' comments:

Reviewer #1 (Remarks to the Author):

The authors developed a system based on the integration of a smartphone and microscopy with ultraviolet surface excitation (MUSE), which is called Pocket MUSE. Due to the localized excitation of UV light, Pocket MUSE can image thick samples without physical sectioning, which can be used to image many types of specimens with minimal sample preparation. The authors have designed their optical module which is attached to the smartphone to enable submicron resolution, revealing individual cells in biological tissues. With this simple design, Pocket MUSE should be useful in providing high-resolution images in resource-limited settings.

The novelty of the work is limited as the optical module design is simple. Also, the integration of the smartphone camera with MUSE is expected. However, as shown by the authors, there is a wide range of applications for Pocket MUSE. The results are also encouraging. Therefore, I believe that the public should be interested in this work. I would recommend for acceptance of publication if the authors can show more scientific elements of the work and address my questions below.

1. In lines 76–80, the authors mentioned using frustrated total internal reflection (TIR) as the light delivery. From the authors' drawing, it is still difficult to understand how the LED light goes. Also, with the divergence of the LED light, what is the portion of light energy that can reach the sample through Frustrated TIR?
2. I thought the sample holder should be a quartz optical window for UV light delivery. However, in lines 148–150, the authors are using the word "glass" instead. Please double check.
3. In general, there are not too many things to question. The reason is that the authors only show a collection of images. However, the usefulness of the images has not been shown, which is the weakness of the work. Therefore, I believe that the authors should, at the least, show one of the killing applications of Pocket MUSE to strengthen the significance of the manuscript.

There are some problems with the writing. The authors should revise accordingly:

1. In line 63, the authors mentioned that "to contribute to blur and background". I believe that it should be "to contribute to blur the background".
2. In line 157, there is "Histology imaging" at the end of the sentence. Please double check.

Reviewer #3 (Remarks to the Author):

The authors demonstrated Pocket MUSE, which is an application of a new exciting imaging technique, MUSE (Microscopy with ultraviolet surface excitation), to another exciting emerging modality, smartphone microscopy. The device demonstrated is extremely compact and seems to work with extremely simple manipulation as shown in their video.

Their work is impressive and noble. I believe this is the first demonstration of MUSE on smartphone microscopy platform. However, there are multiple major flaws in the manuscript. They are serious, and I believe it should not be published without massive revision to address these issues.

1. Incomplete data to validate the imaging quality of pocket MUSE relative to other modalities/standard in the Result section.

In H&E section (Fig.2), pocket MUSE images with/without pseudo H&E color remapping are shown, but no comparison with any of standard technologies (standard H&E, benchtop MUSE, or standard

smartphone microscope) are shown. The quality of the pseudo H&E images are not close to standard H&E slide images at all. With the data showing only what pocket MUSE can do, no comparison statement can be made.

In IHC section (Fig.3), what actually was done is a pocket MUSE imaging of thick brain slice with antibody staining. This is not IHC in commonsense (paraffin slides with specific staining protocol), extremely confusing terminology usage. Furthermore, the processing is radically different from conventional IHC processing, and the appearance is also completely different. It is not possible to compare the images to typical IHC even for those who are familiar with IHC slides. Furthermore authors failed to include enough details of the special staining process in manuscript ("slightly higher concentration with overnight staining" Not specific enough to reproduce the experiment.)

The authors also claim "Compared to conventional bright-field imaging, Pocket MUSE reveals more of (lines 176-177)", but what's actually shown is the comparison of pocket MUSE with and without UV excitation. The images shown as the bright-field images in the manuscript are not close to the quality of conventional bright-field smartphone microscope images because of the difference of tissue preparation and likely different optimization of optics. The statement is clearly wrong, and the result doesn't demonstrate the image quality relative to any of the standard/modalities.

Figure.7, too, does show the comparison of pocket MUSE with and without UV. Their statement "pocket MUSE is better than conventional bright-field imaging" is not demonstrated, and the data doesn't serve as an evidence of their high quality imaging.

2. Baseless statements in the Discussion

The supplementary data is insufficient and not convincing to support the statement in line 223-226. Comparison of costs between preliminary smartphone microscopes and pocket MUSE has to be shown to support the statement at lines 231-232.

The statement (fine alignment is not needed for MUSE because of the difference between MUSE and TIR) is not supported in the supplement note (254-255).

No comparison with pocket MUSE vs histology are shown (line 259-260), and psuedo H&E images are far from good as standard H&E.

In 266, 'the excellent IHC performance' was not shown in the paper as explained above.

3. Lack of conciseness

The manuscript has too much insignificant information, and fails to concisely bring key concepts. For example, in materials and methods, there are many insignificant details described such as the details of sanding. On the other hand, there are several key details such as specs of LEDs (power, driving voltage, current, and spectrum) are not shown or can be found only in supplementary materials. Also description of potential future applications of MUSE at the end of discussion section seems quite lengthy, given that feasibility of those are only minimally supported in the manuscript. (Fig 6-8) Also, there are too much materials in supplemental. The data used to justify key concept in the main text has to be moved to main text. Also information included to supplementary must be wisely selected and concisely edited.

Authors must wisely highlight the key information rather than including every subtle details. It seems to me that many of the supplemental information should be submitted as a separate paper in an engineering journal.(as a personal suggestion)

4. Revise for readers of Communications Biology

The manuscript appeared to be a paper originally drafted for histology/medical journal, but is submitted to this biology journal with minor modification. There are too many of expressions "looking like" authors missed to change from the original draft rather than meant to be (like too often mentioned medical/histology as examples). Please read through again carefully and improve consistency.

The intense optical/electrical engineering parts also will be needed to be simplified for general readers

of Communications Biology.

5. incomplete/inappropriate citations

The choices of references are not optimized. About half of the references are for introducing 'hypothetical' application of the technology, not directly relate to the contents of manuscript at this point or even not imaging. In addition, references of key concepts such as MUSE, RACL and TIR are quite thin relative to those. Authors should review the choices of references and reorganize them.

6. Professional writing to public

Authors should focus on showing results, describing methodology, discussing the findings and hypothesis concluded from evidences, rather than describing baseless comments or author-specific stories. For example, I think the first paragraph of supplemental 1.1 contains some inappropriate sentences to be shown in scientific articles.

In short, the technology is impressive and will be of interest to the readers, but there are a lot of works for authors to do to make the manuscript appropriate as a decent scientific writing.

Manuscript ID: COMMSBIO-20-2668-A

Title: **Pocket MUSE: an affordable, versatile and high-performance fluorescence microscope using a smartphone**

Authors: **Yehe Liu, Andrew M. Rollins, Richard M. Levenson, Farzad Fereidouni, Michael W. Jenkins**

All reviewers found merit in our work but had some concerns before publication. We appreciate the suggestions to improve our paper. We have incorporated all the suggestions into the revised manuscript. Responses are highlighted in red. Changed text is highlighted in red in the manuscript.

Reviewer #1

Critique 1a: In lines 76–80, the authors mentioned using frustrated total internal reflection (TIR) as the light delivery. From the authors' drawing, it is still difficult to understand how the LED light goes.

Response 1a: Lines 76-80 are in the introduction, where we did not feel detailed information was warranted. However, we further clarified the position of the LED in our results section (new lines 148-149) and thoroughly discussed "how the LED light goes" (e.g., positioning against the edge of the optical window) in the supplemental material (**Supplemental Note 1.6**).

To address this concern, we also added some brief introduction and referenced the supplemental material in the original lines 76-80 / new lines 81-83.

Critique 1b: Also, with the divergence of the LED light, what is the portion of light energy that can reach the sample through Frustrated TIR?

Response 1b: We appreciate the important question about the portion of light reaching the sample. We didn't know a practical method to accurately measure the percentage of light reaching the sample in this setup. However, we did provide an in-depth discussion about LED coupling with geometric ray-tracing calculations in the supplementary materials (**Supplemental Note 2.1-2.3**). This discussion should effectively address the reviewer's question.

Briefly, the LED die is tightly held against the optical window's edge. Therefore, although the beam divergence is considerable, the spot size of the beam is just slightly larger than the size of the LED die at the edge of the optical window. A significant amount of light can be coupled into the optical window. The percentage amount of the light coupled into the optical window is dependent on the size of the LED die, the distance between the die and the edge of the optical window, and the angle of the edge of the optical window.

Critique 2: I thought the sample holder should be a quartz optical window for UV light delivery. However, in lines 148–150, the authors are using the word “glass” instead. Please double check.

Response 2: Fused quartz is a common optical glass material, but the term “quartz” alone can be ambiguous since it is also a mineral. To clarify this, we changed the term “quartz” to “fused quartz” in new lines 97, 149, 437 & 438.

Critique 3: In general, there are not too many things to question. The reason is that the authors only show a collection of images. However, the usefulness of the images has not been shown, which is the weakness of the work. Therefore, I believe that the authors should, at the least, show one of the killing applications of Pocket MUSE to strengthen the significance of the manuscript.

Response 3: Pocket MUSE is a device for taking microscopy images. To demonstrate the usefulness of Pocket MUSE images for histology, we added additional data to compare and characterize histology images taken with Pocket MUSE (e.g., new Figure 2, new lines 163-171, 579-585). We have also added references demonstrating the great potential of MUSE systems for preliminary histology imaging and biology education (new lines 284-286, 288). In the discussion of the paper, we also clarified the point that Pocket MUSE could extend these applications to locations where a fluorescence microscope is not available (new lines 286).

Critique 4: In line 63, the authors mentioned that “to contribute to blur and background”. I believe that it should be “to contribute to blur the background”.

Response 4: We corrected the sentence in new lines 62-63 as follows. “Without subsurface signals degrading image contrast, strong optical sectioning is achieved near the sample surface.”

Critique 5: In line 157, there is “Histology imaging” at the end of the sentence. Please double check.

Response 5: This is corrected in the revision.

Reviewer #2

Critique 1a: The most novel aspect of the work is the FTIR illumination scheme. However, its effect on MUSE sectioning (axial resolution and contrast) isn't really discussed. From the original MUSE patents and the following experimental work of Yoshitake et al, the effect of angle of illumination is known to be critical to MUSE image resolution. While the manuscript provides detailed calculations and even ray tracing simulations of the FTIR process, the effect of angle of illumination and what effect it has on the pocket MUSE resolution is less clearly addressed.

Response 1a: We thank the reviewer for bringing up this point. Because the angle of illumination affects the axial resolution and contrast of MUSE imaging, we actually added a full section in the supplemental material (**Supplemental Note 2.7**) covering how frustrated TIR affects axial resolution.

Critique 1b: Further, many (all?) calculations use the visible light refractive index of materials, rather than the (higher) 285nm index, and hence are somewhat inaccurate.

Response 1b: We missed that the RI of quartz is 1.49 instead of 1.47 at 285 nm. We further clarified this point and updated the examples to 1.49 in the supplemental material, including **SF 2.2, 2.3 & 2.5 and Supplemental Note 2.1 & 2.3**.

Critique 1c: It should be simple to calculate the effect on axial resolution of the FTIR illumination and compare to conventional illumination (particularly in light of Yoshitake et al), which would greatly aid in evaluating the advantages of FTIR illumination. However, from some simple calculations (which may be inaccurate), I suspect that the design optimized for coupling efficiency rather than reducing angle of refraction into the sample (frustrated angle of refraction into the sample ranges from ~14 to 90 deg, with most of the refracted angles being near ~70-90 deg). This suggests a tradeoff in resolution/contrast, and possible scope for further optimization.

Response 1c: Unlike in the Yoshitake et al. paper, our optical setup is quite different (e.g., involves a much broader distribution of angles incident upon the sample). We are not aware of a method to accurately calculate/measure the exact axial resolution and optical sectioning of the frustrated TIR illumination because of the broad distribution of angles and sample-specific penetration depth. Although we do not report a predicted optical section thickness with our setup, we did add **Supplemental Note 2.7** to discuss the effects of optical sectioning with frustrated TIR illumination based on the concepts presented in Yoshitake's paper.

The reviewer is correct that the design is optimized for coupling efficiency because the photon budget is limited in this application. We would like to note that the reviewer may have misunderstood that no light can reach the sample at any air-immersion angles (impossible through TIR) based on the angle ranges given in the critique. Considering the broader range of angles delivered by frustrated TIR, the overall effect is between air immersion and water immersion from the paper by Yoshitake et al. This discussion is covered in the new supplemental section, and referenced in **new line 270-273**. We believe the tradeoff in resolution/contrast is possible scope for further optimization in the future.

Critique 2a: Sample mounting only relies on surface tension to the quartz plate.

Would this not result in stability issues, especially when some of your exposure times range in seconds?

Response 2a: In most cases, exposure is $\leq 1/3$ s. For the iPhones, that is the upper limit of the shutter speed. For some samples, 1/15 s or faster shutter speed is sufficient at a relatively low camera gain (e.g., ~ISO 400 for the iPhone 6s). Within less than 250 ms exposure, stability was rarely a problem.

Some cameras in Android phones can expose longer, but that is only necessary when the signal is weak. As the reviewer suggested, we do experience some blurriness when the exposure time ranges in seconds. Avoiding high-frequency movement by supporting the smartphone against the chest or a wall is sufficient to eliminate the stability issues when long exposure times are used. It is also always easy to take a few images and select the one with the least blur. Some of this discussion is in the method/Pocket MUSE imaging section (new line 457-477).

Critique 2b: Would this not result in more surface irregularities (from solid tissue samples, despite best efforts in cutting flat surfaces with basic tools) which might result in the following?

Response 2b: As the reviewer asked, we do see some image artifacts resulting from surface irregularities when the sample is large and rigid (e.g., plant root). However, it is relatively easy to cut a flat surface on these samples with razors. In contrast, soft samples are difficult to cut (e.g., unfixed tissue), but surface tension can flatten the surface effectively. Some difficulty comes from fixed animal tissues. Some mechanical support may be helpful for these types of tissues. This discussion is included in the method section/Pocket MUSE imaging section (new line 460-465).

Critique 2bi: Reduce FTIR coupling and show up as void/dim spots in the image

Response 2bi: Void/dim spots could be caused by large sample sizes. However, we did not observe this problem because there was always water in between the sample and the optical window. Out of focused spots are common with strong surface irregularities (e.g., vascular structures).

Critique 2bii: Even if illuminated, your depth of field given your ranges of RACL apertures and common pixel sizes should range from ~5-10 μm . Won't this give unfocused/blurry regions in images? There was no mention of the evaluation of the optical sectioning thickness (effective penetration of UV excitation) but it's unlikely that it's less than your depth of field.

Response 2bii: F/# of the RACL is ~2, which results in an NA of ~0.24 and an ~10 μm depth of field. As the reviewer suggested, this can cause unfocused/blurry regions in images if the sample is not close to the optical window surface.

According to the Yoshitake et al. paper, optical sectioning thickness for water immersion and air immersion is between 10 μm and 20 μm (285 nm) for histology tissues. Optical sectioning of the frustrated TIR is likely in between this range for similar samples and greater for samples with higher water content (e.g., plant tissues, suspensions, etc.).

As the reviewer suggested, we added some discussion about the depth of field and optical sectioning in **Supplemental Note 2.7**.

Critique 2biii: Is focus stacking used? I did not see a mention of it.

Response 2bii: Focus stacking is not used in this work.

Critique 2c: Why not introduce some compressive force behind the tissue samples?

Response 2C: This is actually an excellent recommendation. Some simple 3D printed mechanics can help, and it will be very useful for many types of rigid and not flat samples. We added this recommendation in the method section (new lines 463-465).

Critique 3a: Some questions about PSF/resolution measurements in Figure 1
Typo on the caption on e-h. Δy should be in mm not in μm

Response 3a: We corrected the typo.

Critique 3b: The simulated RACL on d is different from the one used in i which I assume is used for most of the images due to its better performance (since the simulated PSF for the former RACL is worse than the observed resolution from the latter RACL). Is the latter RACL's optical design available to be simulated and if so, why not simulate the PSF for that? If not, why not show the USAF measurements for the former RACL perhaps to compare theoretical vs observed?

Response 3b: Unfortunately, RACL designs and actual lenses are unlinked. Like the microscope objectives, aspheric camera lens manufacturers do not disclose the lenses' optical designs. They also put suboptimal optical designs in their patent applications. It is almost impossible to find out the optical design of a particular RACL, or get a particular RACL corresponding to a particular patent. The simulation basically serves as a proof of concept for the 1:2 relay configuration and predicts the general behavior of this configuration (e.g., distortion in the field).

Critique 3c: How was the USAF target imaged? Was it through the same quartz plate (which being 500 μm thick should add some spherical aberration contribution)?

Response 3c: The USAF target was imaged through the quartz optical window because almost all of these RACLs are designed to work with a ~ 0.3 mm thick borosilicate IR filter in front of the camera sensor (e.g., see the lower lens in **Fig. 1d**). Spherical aberration will be similar or more significant without the optical window. This is clarified in the **Supplemental Figure 4.1**.

Critique 3d: Would an effective resolution measurement be better? I.e. suspended fluorescent beads to measure effective PSF.

Response 3d: We did a beads-based resolution measurement and the result is included in the supplemental materials (**Supplemental Fig. 4.8**).

Critique 4a: Mentioning the exposure times for the shown images in the figures would be appreciated.

Curious to see how long exposures have to be for corresponding tissue/stain to get acceptable SNR.

Response 4a: We have included a table (**Supplemental Table 4.1**) showing the exposures/ISO of most images in the main manuscript. Generally speaking, 1/10s and an ISO of 400 is a good starting point for the iPhone 6s+.

Critique 4b: Also couples into the sample stability or hand stability if the exposure times are significantly long.

Response 4b: This is discussed above.

Critique 5: From supplementary text: “In addition, while many smartphone camera sensors have 12 or 16 bit pixel depth, image data are almost always converted to an 8 bit lossy RGB JPEG format by the default camera apps. Useful information from high bit depth resolution and dynamic range is lost in the process.”

8 bit JPEG files use the sRGB color space, which is gamma corrected. RAW images are simply the linear sensor values and thus proportional to photon counts. An 8 bit sRGB image has nearly the same dynamic range as a 12 bit linear image, and both are probably excessive given the full well capacities of 1-2 micron pixels. Rather than loss of dynamic range, the advantage of RAW is that you can calibrate the image to get the number of photons per pixel, whereas with JPEG, you have no idea how the image-signal processor is converting to sRGB or what information it may choose to discard. Likely the conversion is adaptive (and thus not necessarily consistent) and paired with extensive image processing that may or may not be beneficial. This could be highly problematic in some applications (e.g. interpretation of diagnostic images).

Response 5: We agree with the comments of the reviewer and have added text to clarify the advantage of the RAW image to get the number of photons per pixel in **Supplemental Note 2.6**.

Critique 6: Regarding Supplemental Figure 3.1, if the RAW data is available, this would be a lot more informative if the sensor gain (photons per DN) was measured (e.g. using EMVA1288 which is simple to do from RAW exposures) and the intensities given in units of photoelectrons rather than RGB values. Then the actual brightness of each fluorophore could be more quantitatively evaluated. This would be valuable because few fluorophores have been imaged at 285nm.

Response 6: For **Supplemental Fig. 3.1**, it is purely for qualitative evaluation. Unfortunately, the images were acquired using a dissection microscope with a different camera. Only Gamma corrected RGB images are saved. The concentration of the dyes on the blots are also not uniformly distributed. It would be interesting to characterize the actual quantum efficiency of the dyes at 285 nm, but this is beyond the scope of the manuscript.

We also found a good database for many common dyes covering spectrum under 300 nm (www.photochemcad.com) This is added to the **Supplemental Note 3.1**.

Critique 7: Typos: Supplementary table of contents 1.5 Should be 3D printed components (missing 3).

Response 7: Typo fixed.

Reviewer #3

Critique 1a: Incomplete data to validate the imaging quality of pocket MUSE relative to other modalities/standard in the Result section. In H&E section (Fig.2), pocket MUSE images with/without pseudo H&E color remapping are shown, but no comparison with any of standard technologies (standard H&E, benchtop MUSE, or standard smartphone microscope) are shown.

Response 1a: As the reviewer asked, we added data acquired with standard H&E and benchtop MUSE systems for reference in the **new Fig. 2 (new lines 163-171)**. Since there is an extremely broad range of smartphone microscopes/samples and there is not a “standard” smartphone microscope, we only provide an example of the special RACL-based bright-field imaging.

Critique 1b: The quality of the pseudo H&E images are not close to standard H&E slide images at all. With the data showing only what pocket MUSE can do, no comparison statement can be made.

Response 1b: We were well aware that the pseudo-H&E images are not the same as standard H&E images. Pseudo-H&E only aims to provide a color reference. Indeed, as discussed in previous MUSE pseudo-H&E publications (citation 16):

“While MUSE captures images in fluorescence, histologists and pathologists are (currently) most comfortable looking at bright-field, H&E-stained specimens... the conversion generates images that can closely approximate the authentic H&E appearance and thus enhances image familiarity.”

The point of showing pseudo-H&E images in this paper is to demonstrate that this well-established MUSE technique could also be applied to Pocket MUSE. Although pseudo-H&E is different from the standard H&E, especially in image details, the remapping aims to help pathologists better understand the structures in the MUSE data. To address the reviewer’s critique, we also showed images from several different image modalities (e.g., benchtop MUSE, Pocket MUSE, benchtop brightfield of conventional H&E, and RACL brightfield of conventional H&E) in the new **Fig. 2** for reference.

Critique 1c: In IHC section (Fig.3), what actually was done is a pocket MUSE imaging of thick brain slice with antibody staining. This is not IHC in commonsense (paraffin slides with specific staining protocol), extremely confusing terminology usage. Furthermore, the processing is radically different from conventional IHC processing, and the appearance is also completely different. It is not possible to compare the images to typical IHC even for those who are familiar with IHC slides. Furthermore authors failed to include enough details of the special staining process in manuscript (“slightly higher concentration with overnight staining” Not specific enough to reproduce the experiment.)

Response 1c: The reviewer suggested that IHC refers to the more standard procedure of staining and imaging a thin tissue slice in commonsense. We used the term IHC in the manuscript because IHC also refers to antibody staining of biological tissue in a broad sense, including thick/wholmount samples (e.g., <https://www.abcam.com/protocols/whole-mount-staining-protocol>, <https://doi.org/10.1177/002215540205000211>).

To address the reviewer's concern, we clarified the ambiguity by specifying that the process refers to whole-mount fluorescent immunohistochemistry staining, which is different from the conventional thin slice-based IHC (**new lines 178 & 281**).

The reviewer also suggested that we failed to include enough details of the staining process in the result. However, a detailed protocol was provided under the method section / whole-mount samples (**original lines 383-391**). To address the reviewer's concern, we further clarified some experimental details in the result section (**new lines 179-181**).

Critique 1d: The authors also claim "Compared to conventional bright-field imaging, Pocket MUSE reveals more of (lines 176-177)", but what's actually shown is the comparison of pocket MUSE with and without UV excitation. The images shown as the bright-field images in the manuscript are not close to the quality of conventional bright-field smartphone microscope images because of the difference of tissue preparation and likely different optimization of optics. The statement is clearly wrong, and the result doesn't demonstrate the image quality relative to any of the standard/modalities.

Response 1d: The reviewer is correct that this claim can be misleading because bright-field imaging on Pocket MUSE is not exactly the same as conventional bright-field imaging. Therefore, we clarified this claim in **new lines 188-191**. We also provided some images from a benchtop bright-field and fluorescence microscope for reference (**Fig. 5**).

Critique 1e: Figure.7, too, does show the comparison of pocket MUSE with and without UV. Their statement "pocket MUSE is better than conventional bright-field imaging" is not demonstrated, and the data doesn't serve as an evidence of their high quality imaging.

Response 1e: Regarding Figure 7, a comparative statement we made was "Compared to bright-field cytology staining, MUSE fluorescence results in a significantly higher contrast between the cell bodies, nuclei and the background." This statement is misleading because "Compared to bright-field cytology staining" is really ambiguous and broad. What we really meant is on the MUSE device, the specific stain has better contrast under MUSE illumination. We clarified that in the **new lines 218-220**.

Critique 2a: Baseless statements in the Discussion

The supplementary data is insufficient and not convincing to support the statement in line 223-226.

Response 2a: As corrected in the last section, we added some comparison data. We also softened the language from "rival image quality of conventional benchtop microscopes" to "produce high-quality images that are comparable to conventional benchtop microscopes at low magnifications" in **new lines 241-242**, and provided reference images in the **new Fig. 2**.

Critique 2b: Comparison of costs between preliminary smartphone microscopes and pocket MUSE has to be shown to support the statement at lines 231-232.

Response 2b: By doing brief research, we realized that there is a broad range of smartphone microscopes varying from <\$1 to >\$100. The quality and functionalities of these smartphone microscopes also vary significantly. Therefore, it is difficult for us to make a fair and reasonable comparison here. We removed the comparative aspect completely in **new line 247**.

Critique 2c: The statement (fine alignment is not needed for MUSE because of the difference between MUSE and TIR) is not supported in the supplement note (254-255).

Response 2c: The paragraph aims to differentiate our frustrated TIR setup from the conventional TIRF setup. The statement was not trying to make a comparison between MUSE and TIR. We further clarified this statement by changing it to "...the UV LED can be positioned next to the optical window without fine angular alignment." in **new lines 269-270**. Reference to **Supplemental Note 1.5-1.6** provides information to describe the optomechanical configuration.

Critique 2d: No comparison with pocket MUSE vs histology are shown (line 259-260), and psuedo H&E images are far from good as standard H&E.

Response 2d: We added some comparison in several figures (see Response 1a).

Critique 2e: In 266, 'the excellent IHC performance' was not shown in the paper as explained above.

Response 2e: We changed the claim to "our demonstrated IHC performance" in **new line 285**.

Critique 3a: Lack of conciseness

The manuscript has too much insignificant information, and fails to concisely bring key concepts. For example, in materials and methods, there are many insignificant details described such as the details of sanding. On the other hand, there are several key details such as specs of LEDs (power, driving voltage, current, and spectrum) are not shown or can be found only in supplementary materials. Also description of potential future applications of MUSE at the end of discussion section seems quite lengthy, given that feasibility of those are only minimally supported in the manuscript. (Fig 6-8) Also, there are too much materials in supplemental. The data used to justify key concept in the main text has to be moved to main text.

Response 3a: We added some key specs about the LEDs in the main manuscript (**new line 102**). We removed a significant amount of text from the discussion especially those regarding potential/future applications (e.g., removed the paragraph in **original lines 287-296**, reorganized **new lines 296-306**).

Regarding the details of sanding, it is actually an important aspect of aligning the focal point of the system. Being able to align the system without fine tools and fine screw mechanisms is challenging, especially in low resources. Since this is not a typical way of performing optical alignment, we would like to keep these technical details in the main manuscript.

Critique 3d: Also information included to supplementary must be wisely selected and concisely edited. Authors must wisely highlight the key information rather than including every subtle details. It seems to me that many of the supplemental information should be submitted as a separate paper in an engineering journal.(as a personal suggestion)

Response 3d: For general readers, the supplemental information might be too much detail, but it is absolutely helpful for people who want to build their own Pocket MUSE device, especially for those who do not have as much fabrication experience as we do. Readers can always avoid the supplemental material if they are not interested in every subtle detail, but it can save builders much time and effort to troubleshoot these potential pitfalls. We also note that the supplemental material is more organized than a typical supplemental section, so readers can quickly find particular information. Therefore, we would like to keep these details in the supplement instead of writing a separate paper.

Critique 4: Revise for readers of Communications Biology

The manuscript appeared to be a paper originally drafted for histology/medical journal, but is submitted to this biology journal with minor modification. There are too many of expressions "looking like" authors missed to change from the original draft rather than meant to be (like too often mentioned medical/histology as examples). Please read through again carefully and improve consistency.

The intense optical/electrical engineering parts also will be needed to be simplified for general readers of Communications Biology.

Response 4: According to the aims and scope of *Communications Biology* (<https://www.nature.com/commsbio/about/aims>), it covers both basic biological and biomedical science. We would argue that histology/medical falls into biomedical science. Also, *Communications Biology* states that it considers "submissions from adjacent research fields where the central advance of the study is of interest to biologists, for example, chemical biology, biophysics and biomedical engineering." We believe the intense biomedical (optical/electrical) engineering sections are vital for those interested in building our device, but we feel the text is also manageable for those just interested in the biology sections. We also found a number of papers related to optical/biomedical engineering in *Communications Biology*. Some examples are listed below:

<https://www.nature.com/articles/s42003-020-01273-w>

<https://www.nature.com/articles/s42003-020-1029-7>

<https://www.nature.com/articles/s42003-020-01299-0>

Critique 5: incomplete/inappropriate citations

The choices of references are not optimized. About half of the references are for introducing 'hypothetical' application of the technology, not directly relate to the contents of manuscript at this point or even not imaging. In addition, references of key concepts such as MUSE, RACL and TIR are quite thin relative to those. Authors should review the choices of references and reorganize them.

Response 5: We increased the number of citations related to MUSE (17, 18 & 33), RACL (21) and TIR (29-31). We also removed some citations in the discussion section as we simplified the part (see **Response 3a**).

Critique 6: Professional writing to public

Authors should focus on showing results, describing methodology, discussing the findings and hypothesis concluded from evidences, rather than describing baseless comments or author-specific stories. For example, I think the first paragraph of supplemental 1.1 contains some inappropriate sentences to be shown in scientific articles.

Response 6: We agree that there are some author-specific storylines and we did a thorough review to make these comments more appropriate (e.g., we removed the inappropriate/less relevant info in **Supplemental Note 1.1**). The language of the manuscript is also improved for professional writing.

REVIEWERS' COMMENTS:

Reviewer #1 (Remarks to the Author):

As mentioned in the previous round of comments, the level and the appropriateness of the scientific writing is low. The authors have tried to address this issue in the revised manuscript. Most of my comments have been addressed with a sufficient level of detail.

Although the supplementary information is not concise enough, causing problems for the audience to understand the most important contents of the text, the authors have tried to arrange the style as a user manual, which is understandable.

Given that PocketMUSE should be of interest to the general public, especially for STEM education, and together with the newly added data to further support the claims, I would recommend this work for publication in *Communications Biology*.

Reviewer #2 (Remarks to the Author):

I am satisfied with the paper revisions and recommend publication.

Reviewer #3 (Remarks to the Author):

This manuscript demonstrates the pocket MUSE system, which is a combination of two emerging technologies, smartphone microscope and MUSE. The manuscript covers from comprehensive details of system design including lens choices and FTIR illumination, to the demonstration of imaging different types of specimens compared with other imaging modalities. The major revisions authors made further improved the comprehensiveness of validation of image quality, and largely improved the readability.

This manuscript will be beneficial for wide range of researchers and various applications. I would suggest publishing this manuscript.

However, although the manuscript largely improved from the original, I'd like to point out number of remaining problems to be addressed.

1. In abstract, submicron resolution is over claiming. The demonstrated resolution is only approaching to 1 μm . In figure 1i, you claim that element 2 is resolved but it won't be considered as resolved in FWHM sense. Even unclear if element 1 is resolved (not far at least though). Please clarify how you determined resolution.

2. The term "effective resolution" is used in the text but not defined. Please define clearly. Also please carefully use the word "resolution" throughout the document. There are several resolutions to be discussed/need to be considered separately/combined (optical? pixel? combined? measured?).

3. Line 55-57 stresses the importance to develop sample preparation protocol for pocket MUSE, but information for the protocol used in this study is distributed throughout the main text and it is difficult to understand what exactly you did. Although the supplemental document describes staining and sample loading in two pages, you should describe the key steps of your sample preparation protocol in concise way in the main text and describe why it's important/advantageous.

4. In the objective lens selection section, please add more details for the system used in this study, such as NA and diameter of lens, pixel resolution, magnification, filter characteristics (specs of RGB filters). Though selection processes are shown, current writing lacks the final design details.
5. In figure 2, please list NA for RACL to clarify difference of optical resolution. Figure 2e-h are nice comparison to assess the optical performance of the Pocket MUSE system compared with conventional microscope, because the sample was already physically sectioned and MUSE wasn't performed. The magnified view of the cell shows the resolution of the pocket MUSE (brightfield) (e) is even superior to that of 20x 0.45NA microscope (h). Please add discussion in the main text that if this makes sense from the optics configuration of pocket MUSE, or other factors (e.g. pixel resolution) also play a role.
6. In line 168, "Using a similar single-dip staining protocol," single-dip staining protocol hasn't been introduced yet in this paper. (related to 3)
7. In figure 3, it's good that information of camera sensor has added. Could you add camera sensor information to other figures too? It is important information to interpret the resolution of images.
8. In 189, double period (typo)
9. Paragraph starting from 199 is a little bit out of place as you have already shown bright field images in earlier figures. Please consider rephrasing/reorganizing.
10. For histology imaging, please elaborate more about the features pocket MUSE is/is not able to visualize. For example, in figure 2, smartphone microscope H&E (2e) shows nucleoli nicely, but nucleoli are hardly identifiable in MUSE images of thick specimens (2b,2d). Please describe limitations and explain why it's limited.
11. In line 438, typo, space missing between quartz and optical.
12. Please add discussion about UV safety for operators.

Title: **Pocket MUSE: an affordable, versatile and high-performance fluorescence microscope using a smartphone**

Authors: **Yehe Liu, Andrew M. Rollins, Richard M. Levenson, Farzad Fereidouni, Michael W. Jenkins**

We appreciate the reviewer's suggestions to improve our paper. We have incorporated all the suggestions into the revised manuscript. Responses are highlighted in red. Changed text is highlighted in red in the manuscript.

1. In abstract, submicron resolution is over claiming. The demonstrated resolution is only approaching to 1 μm . In figure 1i, you claim that element 2 is resolved but it won't be considered as resolved in FWHM sense. Even unclear if element 1 is resolved (not far at least though). Please clarify how you determined resolution.

There is no good standard for interpreting the USAF-1951 test images. Based on either the Rayleigh or the Sparrow (FWHM) resolution criterion, it is fair to claim the resolution of the system is less than a micron (see figures below). In figure 1i element 1&2, the resolution clearly surpass Sparrow criterion. We clarified the criterion in the figure caption.

2. The term “effective resolution” is used in the text but not defined. Please define clearly. Also please carefully use the word “resolution” throughout the document. There are several resolutions to be discussed/need to be considered separately/combined (optical? pixel? combined? measured?).

Effective resolution refers to the resolution limited by either optical or pixel sampling, which is clarified in the last paragraph of the introduction.

3. Line 55-57 stresses the importance to develop sample preparation protocol for pocket MUSE, but information for the protocol used in this study is distributed throughout the main text and it is difficult to understand what exactly you did. Although the supplemental document describes staining and sample loading in two pages, you should describe the key steps of your sample preparation protocol in concise way in the main text and describe why it's important/advantageous.

Line 55-57 aims to prepare/support a point in the upcoming paragraph, where we identified that MUSE is a good supplement to smartphone microscopes because sample preparation with MUSE is simple. We changed the wording in line 55-57 so that the reader does not think we are developing new preparation protocols.

Even though MUSE sample preparation is simple, the preparation still depends on the type of sample. The manuscript is organized in a way that each type of sample is discussed in its own section. We feel this simpler to follow. For solid samples the protocol is "one dipping in a dye" followed by a rinse. For suspensions the protocol is mixing the sample in a dye solution. Some samples do not require any processing and can be imaged directly. The protocols are concisely described in the method section for each type of sample, and elaborated in the supplemental document.

4. In the objective lens selection section, please add more details for the system used in this study, such as NA and diameter of lens, pixel resolution, magnification, filter characteristics (specs of RGB filters). Though selection processes are shown, current writing lacks the final design details.

NA and diameter of the lens, and the camera specs of the phone are added in the section. Specs of RGB filters are Sony's confidential information and therefore are not included.

5. In figure 2, please list NA for RACL to clarify difference of optical resolution. Figure 2e-h are nice comparison to assess the optical performance of the Pocket MUSE system compared with conventional microscope, because the sample was already physically sectioned and MUSE wasn't performed. The magnified view of the cell shows the resolution of the pocket MUSE (brightfield) (e) is even superior to that of 20x 0.45NA microscope (h). Please add discussion in the main text that if this makes sense from the optics configuration of pocket MUSE, or other factors (e.g. pixel resolution) also play a role.

NA of the RACL is discussed in the result section (see comment 4). Discussion is added in the first paragraph of the discussion section.

6. In line 168, "Using a similar single-dip staining protocol," single-dip staining protocol hasn't been introduced yet in this paper. (related to 3)

The "similar single-dip staining protocol" refers to the protocol described in the original MUSE publication. Clarification and citation are added.

7. In figure 3, it's good that information of camera sensor has added. Could you add camera sensor information to other figures too? It is important information to interpret the resolution of images.

We added camera sensor information in the objective lens selection section (see comment 4). In the method section, we also clarified that “most Pocket MUSE images in this manuscript are acquired with the same Pocket MUSE device and an iPhone 6s+ smartphone unless otherwise specified”.

8. In 189, double period (typo)

Typo corrected.

9. Paragraph starting from 199 is a little bit out of place as you have already shown bright field images in earlier figures. Please consider rephrasing/reorganizing.

The bright-field images in earlier figures were taken using an RACL attached to a smartphone, not the assembled Pocket MUSE. The paragraph starting from 199 refers to the assembled Pocket MUSE device. We clarified this point in the paragraph.

10. For histology imaging, please elaborate more about the features pocket MUSE is/is not able to visualize. For example, in figure 2, smartphone microscope H&E (2e) shows nucleoli nicely, but nucleoli are hardly identifiable in MUSE images of thick specimens (2b,2d). Please describe limitations and explain why it's limited.

We added a sentence to the caption of figure 2 describing how our Pocket MUSE images do not appear to visualize nucleoli as well. We feel uncomfortable discussing feature differences in great detail since many factors can contribute to image contrast such as dye concentration. A more complete characterization is beyond the scope of this paper.

11. In line 438, typo, space missing between quartz and optical.

Typo corrected.

12. Please add discussion about UV safety for operators.

Discussion about UV safety is added in the discussion section under “Pocket MUSE Imaging”.